# Learning Fair Graph Representations with Multi-view Information Bottleneck

## Abstract

Graph neural networks (GNNs) excel on relational data by passing messages over node features and structure, but they can amplify training data biases, propagating discriminatory attributes and structural imbalances into unfair outcomes. Many fairness methods treat bias as a single source, ignoring distinct attribute and structural effects and leading to suboptimal fairness and utility trade-offs. To overcome this challenge, we propose FairMIB, a Multi-view information bottleneck framework designed to decompose graphs into feature, structural, and Diffusion Views for mitigating complex biases in GNNs. In particular, the proposed FairMIB employs contrastive learning to maximize cross-view mutual information for bias-free representation learning. It further integrates multi-perspective conditional information bottleneck objectives to balance task utility and fairness by minimizing mutual information with sensitive attributes. Additionally, FairMIB introduces an inverse probability-weighted (IPW) adjacency correction in the Diffusion View, which reduces the spread of bias propagation during message passing. Experiments on five real-world benchmark datasets demonstrate that FairMIB achieves state-of-the-art performance across both utility and fairness metrics.

## 1 Introduction

Graph Neural Networks (GNNs) represent a pivotal advancement in machine learning, offering a powerful paradigm for modeling complex relational data Wu et al. (2020b). Through a message-passing mechanism, GNNs iteratively aggregate information from a node and its neighbors, effectively capturing both node attributes and the graph's structural dependencies Mo et al. (2025). This capability to learn from intricate patterns has established GNNs as indispensable tools in various high-stakes domains, such as recommender systems Amara et al. (2025), drug discovery Wang et al. (2025), and social network analysis Feng & Qian (2025). Across these applications, GNNs consistently deliver superior performance by exploiting the rich interplay between node features and network topology, ultimately leading to more accurate and scalable predictions Chen et al. (2025).

Nevertheless, in real-world applications, data are inherently imperfect Ju et al. (2024b); Zhan et al. (2026), with biases arising from sampling bias, selection bias, and labeling bias Guo et al. (2023); Li et al. (2024b). When trained on such data, GNNs inevitably internalize and even amplify these biases Dai & Wang (2021), leading to outputs that may exhibit discriminatory or unfair behaviors Agarwal et al. (2021). Such unfairness not only compromises the reliability and practical adoption of models but also poses broader societal risks, including the potential erosion of public trust in intelligent systems. For example, biases in risk assessment tools toward specific groups can lead to unfair sentencing and bail decisions Lowden (2018), while systemic biases in credit scoring models against certain regions may result in inequitable loan approvals Li et al. (2024a). Consequently, fairness has emerged as a critical challenge that must be addressed to ensure the trustworthy deployment of Graph Convolutional Networks (GCNs) Zhang et al. (2024c).

Existing studies on fairness in GNNs generally fall into two categories: data-level methods Zhang et al. (2024a); Agarwal et al. (2021) and model-level methods Yang et al. (2024); Lee et al. (2025). The first category corrects biases at the data level through pre-processing techniques that adjust the data distribution, re-balance underrepresented groups, or modify node features and graph structures before training. By addressing bias at the source, these methods aim to mitigate its propagation during model learning. For example, FairGB Li et al. (2024b) achieves re-balancing by introducing

counterfactual node mixup and contribution alignment loss, while FG-SMOTE Wang et al. (2024) creates synthetic nodes for underrepresented groups, assigns sensitive attributes proportionally, and applies fair link prediction to generate non-discriminatory connections, thereby correcting both distributional and structural biases. Model-level approaches, on the other hand, incorporate fairness directly into the training process, typically by embedding fairness as a regularization term or constraint within the objective function. These methods aim to restrict the leakage of sensitive information, ensuring that learned representations remain predictive while minimizing dependence on protected attributes. For example, FairVGNN Wang et al. (2022) enhances fairness by generating fairness-aware Feature Views and applying adaptive weight pruning to mitigate sensitive attribute leakage during feature propagation. Similarly, FairDLA Zhen et al. (2025) decouples task-related and bias-related representations, then performs dual-layer alignment at both the sensitive attribute group level and the task subgroup level to enhance fairness.

Model-based methods generally learn a single node representation where attribute, structural, and propagation biases are intertwined, making it difficult to separate them from task-relevant features and leaving residual sensitive information Zhu et al. (2024). Data-level pre-processing methods can partially alleviate data imbalance Wang et al. (2024), but often treat distributional skew in isolation, overlooking how sensitive information propagates and entangles during message passing Yang et al. (2024). These methods often assume static and accurate graph structures and features, overlooking noise, missing data, and outliers, which cause models to retain latent bias signals and result in incomplete debiasing and fairness gaps Zhou et al. (2020); Li et al. (2024b). Model-based methods typically rely on a single, entangled node representation that conflates bias signals from attributes, structure, and propagation, leading to incomplete debiasing and residual sensitive information Zhu et al. (2024). Such entanglement makes it difficult to disentangle the sources of bias, potentially suppressing task-relevant features while retaining sensitive ones. Moreover, as bias accumulates through multi-layer message passing and representation updates, residual sensitive attributes may still be encoded in the latent space, and fairness constraints alone are often insufficient to prevent systemic bias in downstream tasks Zhen et al. (2025). Furthermore, existing studies largely overlook the issue of cross-view leakage, where biases from attribute-level, structural, and propagation sources interact in complex ways, further amplifying unfair outcomes Lee et al. (2025).

To address these limitations, we propose a novel Multi-view Information Bottleneck framework for Fair GNNs (FairMIB). FairMIB decomposes the graph into distinct informational views: *a Feature View* derived from node attributes, *a Structural View* capturing the pure graph topology, and *a Diffusion View* that models high-order neighborhood information. We employ contrastive learning to maximize mutual information across these views, encouraging the model to learn representations that are invariant to view-specific noise and biases. Concurrently, we integrate the Information Bottleneck (IB) principle as a fairness-aware objective. This objective simultaneously aims to maximize the mutual information between the learned representations and task labels while minimizing the mutual information with sensitive attributes, thereby achieving a principled trade-off between utility and fairness. Our contributions are as follows:

- We propose FairMIB, a novel Multi-view learning framework, designed to decouple and mitigate mixed biases stemming from node attributes and graph structure. The framework decomposes graph data into three independent views: features, structure, and diffusion. It then learns robust node representations by maximizing consistency across the three views.

- We optimize task performance while introducing an IPW based feature matrix correction method in the Diffusion View to block the amplification of sensitive attributes bias during message propagation.

- We perform extensive experiments on five real-world datasets, demonstrating that FairMIB outperforms state-of-the-art baselines in terms of fairness, utility, and stability.

## 2 RELATED WORK

In this section, we briefly review related work, with further details provided in Appendix A. Recent fairness methods for GNNs are typically categorized into pre-processing Rahman et al. (2019); Dong et al. (2022); Li et al. (2024b); Wang et al. (2024), in-processing Dai & Wang (2021); Wang et al. (2022); Agarwal et al. (2021); Yang et al. (2024), and post-processing Lee et al. (2025). These recent works include EDITS Dong et al. (2022) which reweights attributes and perturbs structure

for debiasing, FairGB Li et al. (2024b) which uses resampling and causal contrastive generation to neutralize training views, FairVGNN Wang et al. (2022) which learns channel masks to reduce dependence on sensitive features, NIFTY Agarwal et al. (2021) which employs adversarial and counterfactual augmentations to stabilize embeddings, FairSIN Yang et al. (2024) which injects fairness-promoting features from heterogeneous neighbors before propagation, FairSAD Zhu et al. (2024) which disentangles sensitive factors and applies channel-wise masking, and DAB-GNN Lee et al. (2025) which disentangles attribute and structural bias.

The IB aims to identify a minimal sufficient representation that compresses input data while retaining critical information for subsequent tasks Kawaguchi et al. (2023). This principle has been extended to graph learning through the Graph Information Bottleneck (GIB) model Wu et al. (2020a), which compresses both node features and structural information. In fair graph representation learning, IB shows potential by balancing utility and fairness, such as GRAFair Zhang et al. (2025), which uses a variational graph autoencoder to ensure stable optimization. Recent efforts, such as FDGIB Zheng et al. (2024), combines IB with disentanglement and counterfactual augmentation to decompose node representations into sensitive and non-sensitive subspaces. However, relying on single-view processing is limiting, as graph bias is multi-source, and single representations may conflate signals, leading to under-correction and residual leakage.

## 3 PRELIMINARIES

In this section, we introduce the notations for graph-structured data, followed by a description of commonly used fairness metrics, and then discuss Multi-view information bottleneck and Multi-view conditional information bottleneck (MCIB). More details are presented in Appendix B.

### 3.1 NOTATIONS

We represent an attributed graph as $\mathcal{G} = (\mathcal{V}, \mathcal{E}, \mathbf{X})$, where $\mathcal{V} = \{v_1, v_2, \ldots, v_n\}$ is a set of $n$ nodes, and $\mathcal{E} \subseteq \{(v_i, v_j) | v_i, v_j \in \mathcal{V}\}$ is a set of $m$ edges. The graph's topology is described by the adjacency matrix $\mathbf{A} \in \{0, 1\}^{n \times n}$, where $A_{ij} = 1$ if an edge exists between node $v_i$ and $v_j$, and $0$ otherwise; this definition can be naturally extended to directed or weighted graphs. Each node is associated with features, forming the node feature matrix $\mathbf{X} = [\mathbf{x}_1, \ldots, \mathbf{x}_n]^\top \in \mathbb{R}^{n \times d}$, where $\mathbf{x}_i \in \mathbb{R}^{1 \times d}$ is the $d$-dimensional feature vector for node $v_i$. In the context of fairness research, we use a binary vector $\mathbf{S} \in \{0, 1\}^n$ to represent the sensitive attributes (e.g., gender, race) of all nodes, where $s_i$ is the sensitive attribute value for node $v_i$, which is typically included in the original feature vector $\mathbf{x}_i$. If two nodes $v_u$ and $v_v$ satisfy $s_u = s_v$, they belong to the same demographic group. For the downstream node classification tasks, the ground-truth node labels are represented by the label vector $Y \in \{0, 1\}^n$, while the low-dimensional representations learned by the GNNs form the matrix $\mathbf{Z} \in \mathbb{R}^{n \times d'}$, where $d'$ is the embedding dimension.

### 3.2 MULTI-VIEW INFORMATION BOTTLENECK

From a theoretical perspective, the effectiveness of MIB relies on the redundancy of information across multiple views Cui et al. (2023); Federici et al. (2020). Different views (e.g., $\mathcal{G}_i$ and $\mathcal{G}_j$) often provide overlapping predictive information for the same labels $Y$. In the context of graph data, we formally define view redundancy as follows:

**Definition 1 (View Redundancy)** *A view $\mathcal{G}_i$ is considered redundant with respect to view $\mathcal{G}_j$ for predicting the target labels $Y$ if and only if the mutual information $I(Y; \mathcal{G}_i | \mathcal{G}_j) = 0$. Intuitively, this means that after observing $\mathcal{G}_j$, $\mathcal{G}_i$ adds no new information for predicting $Y$.*

Based on this, the essential objective of MIB is to identify a cross-view minimal sufficient statistic. It aims to learn a highly compressed representation $\mathbf{Z}$ that retains all task-relevant information across the views, making redundant views unnecessary. An ideal, informationally sufficient representation $\mathbf{Z}$ satisfies:

**Corollary 1 (Representation Sufficiency)** *If $\mathbf{Z}$ is a sufficient representation of the views $\{\mathcal{G}_1, \ldots, \mathcal{G}_V\}$, its predictive power for $Y$ is equivalent to that of all views combined:*

$$I(\mathbf{Z}; Y) = I(\mathcal{G}_1, \ldots, \mathcal{G}_V; Y) \tag{1}$$

To achieve this, MIB formulates the learning process as the following optimization problem:

$$\min_{P(\mathbf{Z}|\mathcal{G}_1,\ldots,\mathcal{G}_V)} \sum_{v=1}^{V} I(\mathbf{Z}; \mathcal{G}_v) - \lambda I(\mathbf{Z}; Y) \tag{2}$$

where $I(\mathbf{Z}; \mathcal{G}_v)$ measures the mutual information between the fused representation $\mathbf{Z}$ and a single view $\mathcal{G}_v$, corresponding to the compression objective. $I(\mathbf{Z}; Y)$ measures the mutual information between $\mathbf{Z}$ and the target labels $Y$, representing the relevance to be preserved. The hyperparameter $\lambda$ balances the trade-off between compression and relevance. By solving this, MIB distills the most critical and pure shared knowledge from multiple views for decision-making.

### 3.3 Multi-view Conditional Information Bottleneck (MCIB)

In this section, we introduce the Conditional Fairness Bottleneck (CFB) Gálvez et al. (2021) for fair graph representation learning, and extend it in Multi-view settings. Given views $\{\mathcal{G}_1, \ldots, \mathcal{G}_V\}$, we learn a fair fused representation $\mathbf{Z}$ via the mapping $P(\mathbf{Z} \mid \mathcal{G}_1, \ldots, \mathcal{G}_V)$ that is minimally sufficient for the task while being disentangled from the sensitive attribute $\mathbf{S}$. The goal is to preserve the amount of fair information about the label $Y$ that is independent of $\mathbf{S}$ above a threshold $r$. Formally, the optimization objective of the Multi-view conditional information bottleneck (MCIB) can be defined as:

$$\min_{P(\mathbf{Z}|\mathcal{G}_1,\ldots,\mathcal{G}_V)} \left\{ I(\mathbf{S}; \mathbf{Z}) + \sum_{v=1}^{V} I(\mathcal{G}_v; \mathbf{Z} \mid \mathbf{S}, Y) \right\} \quad \text{s.t.} \quad I(Y; \mathbf{Z} \mid \mathbf{S}) \geq r$$

where $I(\mathbf{S}; \mathbf{Z})$ constrains sensitive information leakage, while the conditional redundancy term $\sum_v I(\mathcal{G}_v; \mathbf{Z} \mid \mathbf{S}, Y)$ eliminates view-specific information that becomes irrelevant once $\mathbf{S}$ and $Y$ are observed. The constraint $I(Y; \mathbf{Z} \mid \mathbf{S}) \geq r$ ensures sufficient task-relevant information is retained, yielding a compact, fair representation that balances utility and fairness across multiple views.

## 4 Methodology

In this section, we present the details of the proposed FairMIB framework. An overview of the framework is provided in Figure 1, which illustrates how it is designed to learn fair node representations from graph data.

### 4.1 Multi-view Disentanglement

Bias in GNNs arises from three main sources: node attributes, graph structure, and the information diffusion mechanism. FairMIB is designed to disentangle these intertwined factors by decomposing them into three complementary views.

#### 4.1.1 Diffusion View

The *Diffusion View* captures potential dynamic deviations that occur as information propagates across the graph. To prevent sensitive attributes from introducing bias during this process, Fair-MIB applies proactive intervention strategies (see Figure 5). Specifically, before propagation, we use IPW Li et al. (2018) to adjust the node feature matrix. The propensity score $e(i)$ represents the probability that a node belongs to the sensitive group, given its features $\mathbf{x}_i$: $e(i) = P(s = 1|\mathbf{x}_i)$. Each node is then assigned a weight based on the IPW formulation:

$$w_i = \frac{s_i}{e(i)} + \frac{1 - s_i}{1 - e(i)} \tag{3}$$

These weights are used to construct a reweighted feature matrix $W = \text{diag}(w_1, \ldots, w_n)$, which produces the debiased feature matrix $\mathbf{X}' = W\mathbf{X}$. This reweighting balances the influence of nodes from different sensitive groups within the feature space.

To model diffusion, we adopt the Personalized Propagation of Neural Predictions (APPNP) Klicpera et al. (2019). A key advantage of APPNP is that it decouples feature transformation from propagation, enabling efficient aggregation of multi-hop neighborhood information. The diffused feature

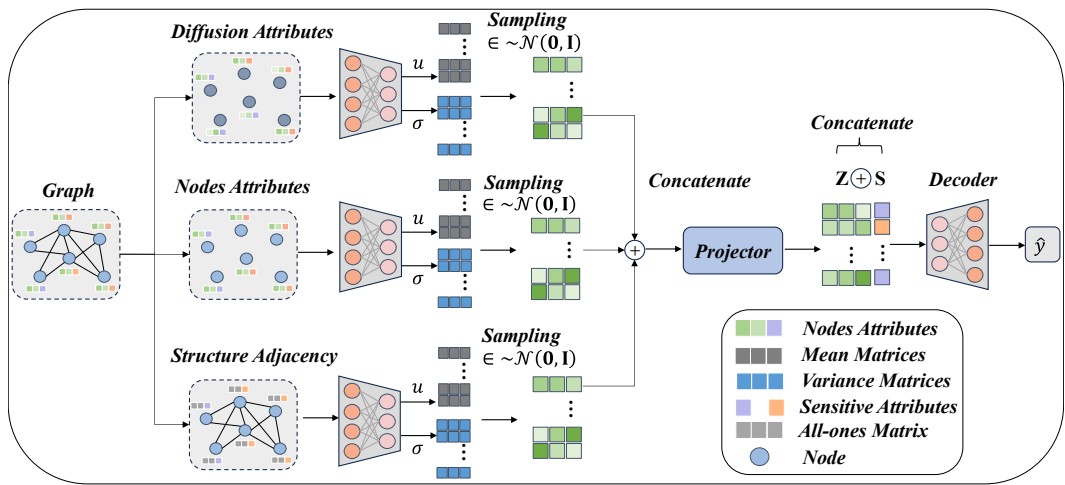

Figure 1: Overview of the proposed FairMIB framework. The model first disentangles the input graph into three complementary views: a Diffusion View, a Feature View, and a Structural View. Each view is encoded by a dedicated variational encoder to obtain a latent representation. These representations are then fused through a Projector, producing a fair representation that is concatenated with the sensitive attribute $\mathbf{S}$ during training to guide the Decoder toward fair predictions.

matrix $\mathbf{X}_{\text{diff}}$ is computed as:

$$\mathbf{X}_{\text{diff}} = \alpha \left( \mathbf{I} - (1 - \alpha) \tilde{\mathbf{A}} \right)^{-1} \mathbf{X}' \tag{4}$$

where $\mathbf{X}'$ is the initial node feature matrix that is propagated along these pathways, and $\alpha$ is the teleport probability that controls the balance between retaining initial features and aggregating neighborhood information.

Finally, the Diffusion View is defined as $\mathcal{G}_{\text{diff}} = (\mathcal{V}, \mathbf{I}, \mathbf{X}_{\text{diff}})$, representing a graph that contains only the debiased node attributes from the fair diffusion process, with structural information implicitly encoded in the features.

### 4.1.2 FEATURE VIEW AND STRUCTURAL VIEW

The *Feature View* isolates the influence of the topological structure, focusing on potential biases in the intrinsic node attributes. It is defined as a graph without inter-node edges, $\mathcal{G}_{\text{feat}} = (\mathcal{V}, \mathbf{I}, \mathbf{X})$, where $\mathbf{X} \in \mathbb{R}^{n \times d}$ is the node feature matrix and $\mathbf{I}$ is the identity matrix, ensuring each node is connected only to itself. This view allows the encoder to learn information solely from the node attributes, isolating biases from structural factors like homophily.

In contrast, the *Structural View* is designed to completely isolate the influence of node attributes, focusing exclusively on the potential biases present within the graph's pure topological structure. It is defined as $\mathcal{G}_{\text{struct}} = (\mathcal{V}, \mathbf{A}, \mathbf{1})$, where $\mathbf{A} \in \mathbb{R}^{n \times n}$ is the original adjacency matrix, and the node feature matrix is replaced by an all-ones matrix $\mathbf{1} \in \mathbb{R}^{n \times d}$. This approach forces the encoder to learn representations solely from connectivity patterns, thereby isolating biases that are introduced by correlations between the node features and sensitive attributes.

### 4.2 FAIR REPRESENTATION LEARNING VIA MCIB

Since the mutual information terms are intractable to optimize directly, we employ variational approximation to derive a tractable objective. For the compression term, $I(\{\mathcal{G}_{\text{view}}\}; \mathbf{Z})$, we use the KL-divergence as its upper bound:

$$I(\{\mathcal{G}_{\text{view}}\}; \mathbf{Z}) \leq \sum_{\text{view}} D_{\text{KL}} \left( p_{\theta_{\text{view}}}(\mathbf{Z}_{\text{view}} | \mathcal{G}_{\text{view}}) \parallel q(\mathbf{Z}_{\text{view}}) \right) \tag{5}$$

where $p_{\theta_{\text{view}}}$ is the posterior distribution defined by a view-specific encoder with parameters $\theta_{\text{view}}$, and $q(\mathbf{Z}_{\text{view}})$ is a prior distribution, typically set to a standard normal distribution $\mathcal{N}(0, \mathbf{I})$.

For the fair prediction term, $I(Y; \mathbf{Z}|\mathbf{S})$, we derive its lower bound:

$$I(Y; \mathbf{Z}|\mathbf{S}) \geq \mathbb{E}_{p(Y,\mathbf{Z},\mathbf{S})}\left[\log p_\phi(Y|\mathbf{Z}, \mathbf{S})\right] \tag{6}$$

where $p_\phi$ is the predictive distribution defined by a decoder with parameters $\phi$.

Combining these bounds, our loss function $\mathcal{L}_{\text{MCFB}}$ can be expressed as:

$$\mathcal{L}_{\text{MCFB}} = \sum_{\text{view}} D_{\text{KL}}(p_{\theta_{\text{view}}}(\mathbf{Z}_{\text{view}} \mid \mathcal{G}_{\text{view}}) \parallel q(\mathbf{Z}_{\text{view}})) - \gamma \, \mathbb{E}_{p(Y,\mathbf{Z},\mathbf{S})}[\log p_\phi(Y \mid \mathbf{Z}, \mathbf{S})] \tag{7}$$

Minimizing this loss function is equivalent to maximizing the Evidence Lower Bound (ELBO), which allows us to achieve our optimization objective in a stable, non-adversarial manner.

### 4.3 MULTI-VIEW CONSISTENCY CONSTRAINT

Although our framework disentangles sources of bias and debiases fused view with a conditional information bottleneck, all three views originate from the same graph, so their fair representations should share a unified task-relevant core in the latent space. To enforce this, we add a Multi-view consistency constraint via contrastive learning Ju et al. (2024a), pulling together a node's debiased representations from different views as positives and pushing apart different nodes as negatives, which drives view-specific encoders to learn a shared, robust, and sensitive invariant semantic space.

We implement this constraint using the InfoNCE loss Rusak et al. (2025). For a node $v_i$, let its latent representations from the feature, structural, and Diffusion Views be $\mathbf{z}_{i,\text{feat}}$, $\mathbf{z}_{i,\text{struct}}$, and $\mathbf{z}_{i,\text{diff}}$. We can select the representations from any two views (e.g., the feature and Structural Views) to form a positive pair $(\mathbf{z}_{i,\text{feat}}, \mathbf{z}_{i,\text{struct}})$. The contrastive loss for this pair is:

$$\mathcal{L}_{\text{con}}(\mathbf{z}_{i,\text{feat}}, \mathbf{z}_{i,\text{struct}}) = -\log \frac{\exp(\text{sim}(\mathbf{z}_{i,\text{feat}}, \mathbf{z}_{i,\text{struct}})/\tau)}{\sum_{j=1}^{N} \exp(\text{sim}(\mathbf{z}_{i,\text{feat}}, \mathbf{z}_{j,\text{struct}})/\tau)} \tag{8}$$

where $\text{sim}(\mathbf{u}, \mathbf{v})$ is a function that measures the similarity between two vectors, typically cosine similarity. The term $\tau$ is a temperature hyperparameter that adjusts the distribution of the similarity scores, and $N$ is the total number of nodes in the batch. The denominator includes the similarity scores between the anchor $\mathbf{z}_{i,\text{feat}}$ and one positive sample $\mathbf{z}_{i,\text{struct}}$, as well as $N-1$ negative samples $\mathbf{z}_{j,\text{struct}}$ for $j \neq i$.

We apply this loss function to all pairwise combinations of the views and average the result over all nodes to obtain the final Multi-view consistency loss $\mathcal{L}_{\text{con}}$:

$$\mathcal{L}_{\text{con}} = \frac{1}{N} \sum_{i=1}^{N} \Big( \mathcal{L}_{\text{con}}(\mathbf{z}_{i,\text{feat}}, \mathbf{z}_{i,\text{struct}}) + \mathcal{L}_{\text{con}}(\mathbf{z}_{i,\text{feat}}, \mathbf{z}_{i,\text{diff}}) + \mathcal{L}_{\text{con}}(\mathbf{z}_{i,\text{struct}}, \mathbf{z}_{i,\text{diff}}) \Big) \tag{9}$$

By minimizing $\mathcal{L}_{\text{con}}$, the model encourages the representations of the same node across different views to be close, while separating representations of different nodes. This promotes the learning of a well-structured, semantically consistent, and fair representation space.

### 4.4 THE OBJECTIVE FUNCTION OF FAIRMIB METHOD

During training, we concatenate the projected representation $\mathbf{Z}_{\text{proj}}$ with the ground-truth sensitive attributes $\mathbf{S}$. This combined vector is then fed into a decoder, $h_\phi$, implemented as a multilayer perceptron (MLP), to make the final node classification predictions $\hat{\mathbf{y}}$:

$$\hat{\mathbf{y}} = h_\phi([\mathbf{Z}_{\text{proj}} \| \mathbf{S}]) \tag{10}$$

This architectural design compels the encoders and the projector to learn information that remains useful for predicting $Y$ even when $\mathbf{S}$ is provided. Consequently, it encourages the model to ignore spurious correlations that are associated with $\mathbf{S}$ but are irrelevant to the prediction task. The standard cross-entropy loss for the node classification task is defined as:

$$\mathcal{L}_{\text{task}} = -\frac{1}{|\mathcal{V}|} \sum_{v \in \mathcal{V}} (y \log \hat{y} + (1-y) \log(1-\hat{y})) \tag{11}$$

The overall training loss function of our FairMIB combines three main components: the task loss, the Multi-view conditional fairness bottleneck loss, and the Multi-view consistency loss. The total loss is given by:

$$\mathcal{L}_{\text{total}} = \mathcal{L}_{\text{task}} + \lambda_{\text{KL}}\mathcal{L}_{\text{MCFB}} + \lambda_{\text{con}}\mathcal{L}_{\text{con}} \tag{12}$$

where $\lambda_{\text{KL}}$ and $\lambda_{\text{con}}$ are hyperparameters that balance the objectives of information compression and cross-view consistency, respectively.

## 5 EXPERIMENTS

In this section, we evaluate the proposed FairMIB framework on five real-world graph datasets. More details of datasets, compared methods, experimental settings, experimental results and analysis are provided in Appendix D due to page limitation. Our evaluation is guided by the following research questions:

**RQ1**: Does FairMIB achieve superior performance in both utility and fairness compared with state-of-the-art baselines? **RQ2**: What is the contribution of each component within the proposed FairMIB framework to overall performance? **RQ3**: How do different informational views (feature, structural, diffusion) affect representation quality and fairness outcomes? **RQ4**: How sensitive is FairMIB to variations in hyperparameter settings?

### 5.1 EXPERIMENTAL SETTINGS

#### 5.1.1 DATASETS AND EVALUATION METRICS

We conducted experiments on five widely used benchmark datasets: German Asuncion & Newman (2007), Bail Jordan & Freiburger (2015), Credit Yeh & Lien (2009), Pokec-z, and Pokec-n Takac & Zabovsky (2012). For model effectiveness, we assess node classification performance using accuracy, F1-score, and AUC-ROC. To evaluate fairness, we adopt Demographic Parity (DP) and Equal Opportunity (EO) as metrics (Appendix B), where lower values indicate higher levels of fairness.

#### 5.1.2 BASELINES

We benchmarked the proposed method against seven state-of-the-art (SOTA) approaches for fair node representation learning, including adversarial methods FairGNN Dai & Wang (2021) and FairVGNN Wang et al. (2022), data augmentation-based methods NIFTY Agarwal et al. (2021), EDITS Dong et al. (2022), and FairGB Li et al. (2024b) , an information bottleneck-based method GRAFair Zhang et al. (2025), and a disentangled representation learning method DAB-GNN Lee et al. (2025).

#### 5.1.3 IMPLEMENTATION DETAILS

For the German, Bail, Credit, and Pokec datasets, we followed the training, validation, and test set splitting scheme proposed in Li et al. (2024b); Yang et al. (2024). For all comparison methods, model hyperparameters were either set according to their official implementations or tuned via grid search to ensure fairness. All models were optimized using the *Adam* optimizer Kingma & Ba (2015), with early stopping based on the validation loss. Following Zhang et al. (2024b), the number of hops in the Diffusion View was fixed at $K = 3$. To ensure robustness, we report the mean and standard deviation over five independent runs with different random seeds. All experiments were conducted on an NVIDIA GeForce GTX 4060 GPU (8 GB).

### 5.2 RQ1: PERFORMANCE COMPARISON

We conducted comprehensive experiments on five benchmark datasets, comparing the proposed FairMIB with a standard GCN baseline and seven state-of-the-art fairness-aware methods. The results in Table 1 highlight the following key findings: (1) The proposed FairMIB framework consistently outperforms the SOTA baselines in terms of fairness while maintaining competitive utility. For example, on the German dataset, the proposed method reduces DP and EO by 98.8% and 99.3% relative to GCN, and its EO is 75% lower than the best-performing baseline GRAFair. On the

Table 1: Comparison of utility and fairness performance across different GNNs fairness methods on five datasets. The datasets are represented as follows: **I** (German), **II** (Bail), **III** (Credit), **IV** (Pokec-z), and **V** (Pokec-n). Arrow (↑) indicates that higher values are better, while (↓) indicates that lower values are better.

| | Metrics | Model | | | | | | | | |
|---|---|---|---|---|---|---|---|---|---|---|
| | | Vanilla GCN | NIFTY | EDITS | FairGNN | FairVGNN | FairGB | GRAFair | DAB-GNN | FairMIB |
| **I** | AUC (↑) | **73.49 ± 2.15** | 68.78 ± 2.69 | 69.41 ± 2.33 | 67.35 ± 2.13 | 72.12 ± 1.10 | 59.77 ± 7.59 | 70.32 ± 1.12 | 66.59 ± 4.30 | 65.55 ± 1.61 |
| | F1 (↑) | 80.76 ± 2.35 | 81.40 ± 0.50 | 81.55 ± 0.59 | 82.01 ± 0.26 | 82.14 ± 0.42 | **82.46 ± 0.23** | 81.95 ± 0.33 | 82.16 ± 0.33 | 82.45 ± 0.20 |
| | ACC (↑) | **71.04 ± 2.36** | 69.92 ± 1.14 | 70.22 ± 0.89 | 69.68 ± 0.30 | 70.16 ± 0.86 | 70.01 ± 0.73 | 70.06 ± 0.16 | 70.12 ± 0.63 | 70.24 ± 0.48 |
| | DP (↓) | 33.75 ± 12.34 | 5.73 ± 5.25 | 4.05 ± 4.48 | 3.49 ± 2.15 | 1.68 ± 0.98 | 1.68 ± 3.30 | 0.91 ± 0.47 | 1.19 ± 1.25 | **0.38 ± 0.76** |
| | EO (↓) | 25.73 ± 8.36 | 5.08 ± 4.29 | 3.89 ± 4.23 | 3.40 ± 2.15 | 1.21 ± 2.11 | 1.08 ± 1.80 | 0.68 ± 0.56 | 1.18 ± 1.75 | **0.17 ± 0.34** |
| **II** | AUC (↑) | 87.39 ± 0.17 | 78.20 ± 2.78 | 86.44 ± 2.17 | 87.36 ± 0.90 | 85.68 ± 0.37 | 87.68 ± 1.41 | 88.68 ± 1.35 | 89.08 ± 3.34 | **89.18 ± 2.15** |
| | F1 (↑) | 77.63 ± 0.42 | 64.76 ± 3.91 | 75.58 ± 3.77 | 77.50 ± 1.69 | 79.11 ± 0.33 | 77.08 ± 2.00 | 80.03 ± 0.56 | 79.79 ± 2.02 | **80.10 ± 1.25** |
| | ACC (↑) | 82.58 ± 1.21 | 74.19 ± 2.57 | 84.49 ± 2.27 | 82.94 ± 1.67 | 84.73 ± 0.46 | 83.31 ± 1.90 | 83.97 ± 1.90 | **89.73 ± 1.02** | 85.62 ± 0.81 |
| | DP (↓) | 6.94 ± 0.21 | 2.44 ± 1.29 | 6.64 ± 0.39 | 6.90 ± 0.17 | 6.53 ± 0.67 | 5.17 ± 0.36 | 1.32 ± 0.43 | **0.92 ± 0.53** | 1.23 ± 0.49 |
| | EO (↓) | 5.56 ± 0.37 | 1.72 ± 1.08 | 7.51 ± 1.20 | 4.65 ± 0.14 | 4.95 ± 1.22 | 3.44 ± 1.20 | 1.46 ± 0.28 | 1.26 ± 0.38 | **1.17 ± 0.45** |
| **III** | AUC (↑) | 72.80 ± 0.23 | 71.96 ± 0.19 | 73.01 ± 0.11 | 71.95 ± 1.43 | 71.34 ± 0.41 | 73.21 ± 0.83 | 72.04 ± 0.42 | 71.34 ± 0.76 | **73.49 ± 0.51** |
| | F1 (↑) | 82.93 ± 0.21 | 81.72 ± 0.05 | 81.81 ± 0.28 | 81.84 ± 1.19 | 87.08 ± 0.74 | 85.83 ± 3.34 | 87.44 ± 0.23 | 87.28 ± 1.06 | **87.79 ± 0.27** |
| | ACC (↑) | 73.99 ± 0.01 | 73.45 ± 0.06 | 73.51 ± 0.30 | 73.41 ± 1.24 | 78.04 ± 0.33 | 77.54 ± 3.48 | 77.34 ± 1.43 | 78.28 ± 1.37 | **78.57 ± 0.86** |
| | DP (↓) | 12.53 ± 0.25 | 11.68 ± 0.07 | 10.90 ± 1.22 | 12.64 ± 2.11 | 5.02 ± 5.22 | 2.30 ± 3.00 | 1.06 ± 0.71 | 0.67 ± 0.76 | **0.40 ± 0.69** |
| | EO (↓) | 10.63 ± 0.02 | 9.39 ± 0.07 | 8.75 ± 1.21 | 10.41 ± 2.03 | 3.60 ± 4.31 | 1.75 ± 2.07 | 0.64 ± 0.26 | 0.49 ± 0.68 | **0.24 ± 0.48** |
| **IV** | AUC (↑) | 72.42 ± 0.33 | 71.59 ± 0.17 | OOM | 73.12 ± 0.12 | **76.02 ± 0.16** | OOM | 69.11 ± 2.27 | 72.02 ± 0.22 | 73.15 ± 1.64 |
| | F1 (↑) | 70.32 ± 0.20 | 67.13 ± 1.66 | OOM | 67.65 ± 1.65 | **70.45 ± 0.57** | OOM | 64.21 ± 1.53 | 64.57 ± 1.76 | 68.86 ± 1.40 |
| | ACC (↑) | **68.54 ± 0.32** | 66.24 ± 0.34 | OOM | 66.24 ± 0.34 | 68.24 ± 0.17 | OOM | 62.29 ± 0.17 | 67.34 ± 1.33 | 66.16 ± 1.43 |
| | DP (↓) | 4.21 ± 0.32 | 6.50 ± 2.16 | OOM | 2.73 ± 2.23 | 2.90 ± 0.77 | OOM | 1.41 ± 1.73 | 1.55 ± 0.45 | **0.69 ± 0.26** |
| | EO (↓) | 4.29 ± 0.24 | 6.43 ± 1.73 | OOM | 2.17 ± 1.85 | 3.09 ± 0.97 | OOM | 1.73 ± 1.49 | 1.12 ± 0.77 | **0.52 ± 0.38** |
| **V** | AUC (↑) | 72.12 ± 0.57 | 71.28 ± 0.35 | OOM | 71.49 ± 0.28 | 73.22 ± 0.92 | OOM | 72.28 ± 1.46 | 73.11 ± 0.36 | **73.28 ± 1.02** |
| | F1 (↑) | 66.78 ± 1.09 | 64.02 ± 1.26 | OOM | 64.80 ± 0.89 | 63.35 ± 1.64 | OOM | 65.75 ± 1.71 | 68.62 ± 1.22 | **69.02 ± 1.91** |
| | ACC (↑) | 66.22 ± 1.09 | 66.14 ± 0.49 | OOM | 65.36 ± 2.06 | 66.16 ± 0.72 | OOM | 66.21 ± 2.35 | **67.23 ± 0.81** | 66.37 ± 1.09 |
| | DP (↓) | 2.83 ± 0.46 | 1.62 ± 0.94 | OOM | 2.26 ± 1.19 | 4.28 ± 1.33 | OOM | 3.22 ± 1.18 | 1.57 ± 0.73 | **1.12 ± 0.76** |
| | EO (↓) | 3.66 ± 0.43 | 1.83 ± 1.12 | OOM | 3.21 ± 2.28 | 5.34 ± 1.27 | OOM | 2.65 ± 1.07 | 1.36 ± 0.54 | **0.92 ± 0.90** |

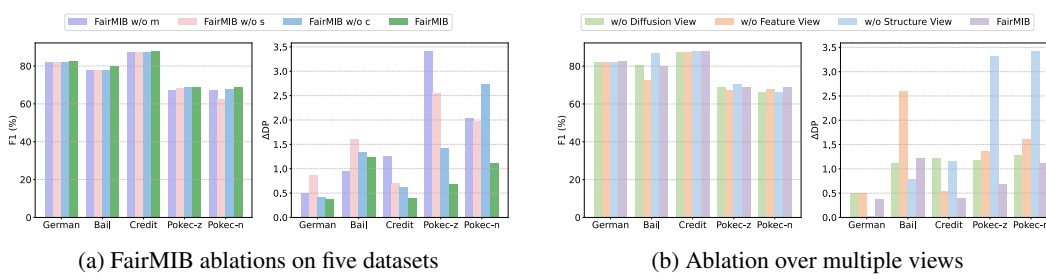

(a) FairMIB ablations on five datasets    (b) Ablation over multiple views

Figure 2: Ablation study and multi-view study on FairMIB

Bail dataset, FairMIB achieves an EO reduction of 82.3% over GCN, while improving F1-score by 3.2%, surpassing all other fairness-aware methods. (2) The proposed FairMIB framework demonstrates strong scalability and utility preservation on large-scale datasets. For instance, on the Pokec-n dataset, it improves the F1-score by 3.4% over GCN and achieves the best fairness, with an EO value 32.4% lower than the runner-up model, DAB-GNN. Similarly, on the Pokec-z dataset, its DP and EO metrics are 51.1% and 53.6% lower than the strongest competitors, respectively. These results confirm the superior balance and scalability of our approach on large-scale graphs.

## 5.3 RQ2: ABLATION STUDY

To answer RQ2, we conducted ablations on our FairMIB with three variants: FairMIB w/o m (removing information compression), FairMIB w/o s (removing the conditional constraint), and FairMIB w/o c (removing Multi-view consistency). As shown in Figure 2a, removing the conditional module (w/o s) significantly degrades fairness across datasets; for example, DP worsens by over 30% on Bail, confirming the need to maximize $I(\mathbf{Y}; \mathbf{Z} \mid \mathbf{S})$ by conditioning on $\mathbf{S}$. Removing compression (w/o m), which minimizes $I(\{\mathcal{G}_{\text{view}}\}; \mathbf{Z})$, harms both utility and fairness, most notably on Pokec-z where DP nearly quadruples, showing that filtering redundant information improves both. Removing consistency (w/o c) also reduces fairness, especially on Pokec-n where DP and EO are worse than other variants, indicating that contrastive alignment of view specific representations yields a robust shared latent space. Overall, these studies verify that the conditional bottleneck, compression, and Multi-view consistency work together to mitigate sensitive attribute bias by enforcing the fairness objective, filtering irrelevant information, and aligning Multi-view semantics.

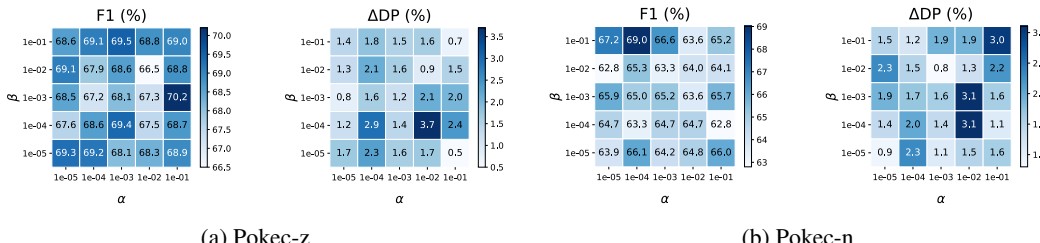

(a) Pokec-z

(b) Pokec-n

Figure 3: Parameter sensitivity results on Pokec datasets. Results demonstrate that FairMIB achieves stable performance across a wide range of parameter settings.

### 5.4 RQ3: MULTI-VIEW ANALYSIS

To answer RQ3, we conducted ablation experiments by removing the Diffusion View (w/o Diffusion View), Feature View (w/o Feature View), or Structural View (w/o Structure View). The results in Figure 2b show that the three views provide complementary information, and their combination is essential for balancing utility and fairness. Removing any view typically leads to a significant performance drop. For example, on the Bail dataset, removing the Feature View causes a 10% drop in F1-score and deterioration in DP and EO metrics, indicating the importance of node attributes for fair decision-making. The relative importance of the Structural and Diffusion Views varies across datasets. On the Pokec-z and Pokec-n datasets, removing the Structural View worsens DP by over 380% and 205%, respectively, showing that structural bias is critical in these topologies. In contrast, on the Credit dataset, removing the Diffusion View has the largest negative impact on fairness, increasing DP by 205%, highlighting the role of multi-hop information propagation in bias correction. These results demonstrate that no single view is universally dominant, validating the necessity of our Multi-view decoupling framework.

### 5.5 RQ4: HYPER-PARAMETER SENSITIVITY ANALYSIS

To address RQ4, we perform a sensitivity analysis of FairMIB with respect to two key hyperparameters, $\alpha$ and $\beta$, which control the relative contributions of information compression and view alignment, respectively. Specifically, we evaluate the model by varying $\alpha$ and $\beta$ across the set $10^{-1}, 10^{-2}, 10^{-3}, 10^{-4}, 10^{-5}$ on the Bail, Credit, Pokec-z, and Pokec-n datasets. The results, shown in Figure 3, indicate that FairMIB exhibits robust performance across a wide range of these hyperparameter values. However, setting $\alpha$ and $\beta$ excessively high can lead to performance degradation due to over-compression of information and overly strict enforcement of view alignment. These findings underscore the importance of balancing the two components, suggesting that selecting $\alpha$ and $\beta$ within the range of $10^{-3}$ to $10^{-5}$ achieves a trade-off between utility and fairness.

### 5.6 EFFICIENCY ANALYSIS

In terms of time and space complexity, the main computational cost of FairMIB comes from the APPNP-based feature diffusion and the forward and backward passes of the three encoders. For each training epoch, applying $K$-step APPNP on the weighted features over the graph $\mathcal{G} = (\mathcal{V}, \mathcal{E}, \mathbf{X})$ has a computational complexity of approximately $\mathcal{O}(K(m + n)h)$, where $n = |\mathcal{V}|$ is the number of nodes, $m = |\mathcal{E}|$ is the number of edges, and $h$ denotes the hidden dimension. The three encoders process the raw features $\mathbf{X}$, the diffused features, and the all-one features, respectively. Their computational cost grows linearly with the number of nodes and edges, approximately $\mathcal{O}(ndh + mh)$. Adding the computation for the contrastive loss, the KL regularization, and the classifier, which together require $\mathcal{O}(nh)$, the overall complexity remains nearly linear with respect to graph size. When considering $R$ independent runs and $T$ training epochs, the total time complexity of FairMIB becomes $\mathcal{O}(RT(Kmh + ndh))$. The memory complexity is $\mathcal{O}(n(d + 3h) + m)$, which is on the same order as standard multi-branch GNNs. The propensity score model is pre-trained separately for about 100 steps before the main training process, and this one-time cost is negligible compared with the full training procedure. As shown in Figure 4, we compare the actual running times of different fair graph learning methods under the same settings. The results demonstrate that, while

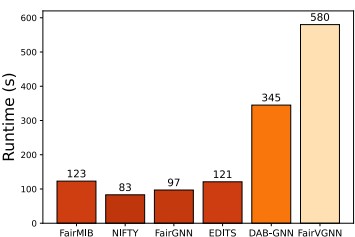

Figure 4: Comparison of Time costs for FairMIB and baselines on Bail

achieving Multi-view representation learning, FairMIB maintains linear scalability comparable to standard graph neural networks and is more efficient than several more complex fair GNN models.

## 5.7 Why Choose APPNP

We adopt APPNP to model higher-order neighborhood effects in a stable and efficient way. We conduct an ablation study on the Bail dataset with five propagation operators: GCN, APPNP, without IPW, SGC, and GCNII in Table 2. APPNP provides the best trade-off among utility, fairness, and efficiency. Compared with GCN, APPNP increases AUC by about 1% and F1 by roughly 1.5%, while reducing training time by over 5% and slightly lowering both fairness gaps (around 10–15% smaller DP and about 7% smaller EO). SGC is roughly 7% faster than APPNP, but suffers from about 2% lower AUC and more than 25% DP gap, indicating that overly aggressive simplification harms fair representation learning. GCNII offers less than 1% additional AUC over APPNP but requires approximately 17% more training time and a considerably more complex architecture. To isolate the effect of the IPW module, we compare APPNP with and without IPW. Introducing IPW reduces the demographic parity gap by nearly 30% and the equal opportunity gap by about 20%, showing that IPW effectively mitigates bias accumulated during diffusion. Overall, APPNP with IPW forms a principled compromise that balances accuracy, fairness, and efficiency while remaining modular.

Table 2: Ablation study for choosing APPNP

| Metric | APPNP | GCN | SGC | GCNII | Without IPW |
|--------|-------|-----|-----|-------|-------------|
| AUC | $88.52 \pm 1.45$ | $87.60 \pm 2.11$ | $86.88 \pm 1.92$ | $89.09 \pm 2.09$ | $88.26 \pm 0.93$ |
| ACC | $84.48 \pm 1.53$ | $83.50 \pm 2.43$ | $84.03 \pm 1.47$ | $85.49 \pm 3.15$ | $84.54 \pm 1.46$ |
| F1 | $78.89 \pm 1.91$ | $77.62 \pm 3.11$ | $76.61 \pm 2.27$ | $80.27 \pm 3.69$ | $78.91 \pm 1.36$ |
| DP | $1.35 \pm 1.23$ | $1.53 \pm 0.84$ | $1.72 \pm 1.45$ | $1.31 \pm 0.35$ | $1.90 \pm 0.75$ |
| EO | $1.39 \pm 0.68$ | $1.49 \pm 0.66$ | $2.86 \pm 2.01$ | $1.39 \pm 0.49$ | $1.72 \pm 0.74$ |
| Time (s) | 123.4915 | 130.2052 | 115.1629 | 144.9692 | 120.7191 |

## 6 Conclusion

This paper addresses the challenge of bias in GNNs from a fairness perspective originating from multi-source information. Traditional approaches often fail to disentangle distinct sources of bias, leading to a suboptimal trade-off between model utility and fairness. To overcome this, we propose FairMIB, a novel framework grounded in the Multi-view conditional information bottleneck principle. Our FairMIB method first disentangles composite graph data into independent feature, structural, and Diffusion Views. It then applies a conditional information bottleneck to the fusion representation to learn compressed representations that preserve task-relevant information while mitigating sensitive attribute leakage. Furthermore, we introduce a Multi-view consistency constraint to ensure semantic alignment across the learned representations. Extensive experiments on five benchmark datasets demonstrate that FairMIB consistently outperforms state-of-the-art methods, achieving a superior balance between fairness and utility. While these results are promising, several avenues for future work remain. The current framework could be extended to more complex scenarios involving multiple intersecting sensitive attributes or enhanced by exploring more diverse strategies for view generation.

## 7 ETHICS STATEMENT

This work investigates fair learning on graphs and proposes a Multi-view conditional information bottleneck for mitigating bias. We use only publicly available datasets under their licenses and do not collect new human subject data. Sensitive attributes are used only during training to encourage conditional fairness and are not required at inference time. We evaluate demographic parity and equality of opportunity, but fairness is context dependent and our results do not guarantee fairness in all deployments. Practitioners should verify consent and data provenance, apply privacy safeguards, conduct domain-specific audits with affected stakeholders, and avoid presenting improvements on chosen metrics as proof of overall neutrality.

## 8 REPRODUCIBILITY STATEMENT

We describe all model components, objectives, and training protocols, including architectures, losses, data preprocessing, and evaluation metrics. We will release code, configuration files, and experiment scripts that reproduce main tables, ablations, and sensitivity analyses with fixed random seeds, reported means and standard deviations over multiple runs, and the exact data splits used. The repository will include a dependency file with package versions, instructions for environment setup, and commands for end-to-end execution on commodity GPUs, enabling independent verification and extension of our results. Our implementation has been submitted in OpenReview and the code will be made publicly available on GitHub.

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

# Appendix

This is the appendix to the paper 'Learning Fair Graph Representations with Multi-view Information Bottleneck'. This appendix provides additional details on related work, preliminaries, the proposed method, and extended experimental results.

## A  RELATED WORK

### A.1  FAIRNESS IN GNNS

In recent years, research on fairness in GNNs has accelerated, with methods commonly grouped into two categories: pre-processing Li et al. (2024b); Dong et al. (2022) and in-processing Yang et al. (2024); Agarwal et al. (2021); Wang et al. (2022). Pre-processing methods address fairness at the data level by rebalancing attribute distributions or modifying graph structures before model training. These methods aim to reduce the unfairness induced by distributional disparities and structural homophily. For example, EDITS Dong et al. (2022) introduces a debiasing framework that jointly optimizes attribute reweighting and structural perturbation in order to reduce attribute and structural bias in graph data. More recently, FairGB Li et al. (2024b) approaches the problem from the perspective of data generation and sampling by combining resampling with causally inspired contrastive generation. This method not only alleviates group bias caused by imbalance in the training set but also provides a more neutral training view for subsequent model learning. A common characteristic of these approaches is the direct modification of input distributions or graph connectivity patterns, thereby ensuring that any downstream GNN can be trained on a relatively fair dataset.

In-processing methods, in contrast, introduce fairness constraints or architectural designs during model learning to suppress the leakage of sensitive information in the message-passing stage Wang et al. (2022); Agarwal et al. (2021); Yang et al. (2024); Zhu et al. (2024); Lee et al. (2025). Since feature propagation can transform channels originally uncorrelated with sensitive attributes into biased ones,

many approaches aim to limit the reliance of propagation channels on sensitive cues. For example, FairVGNN Wang et al. (2022) leverages correlations before and after propagation to learn channel masks that reduce the dependence on sensitive features. NIFTY Agarwal et al. (2021) employs adversarial and counterfactual augmentations to stabilize embeddings and mitigate group separability. FairSIN Yang et al. (2024) proposes a neutralization paradigm that constructs and injects fairness-promoting features from heterogeneous neighbors prior to message passing, thereby offsetting sensitive bias signals and supplementing non-sensitive information.

More recently, disentanglement-based approaches have gained attention. Fair-SAD Zhu et al. (2024) disentangles sensitive-related information into independent components in the representation space and applies channel-wise masking to de-correlate them, thus enhancing fairness while preserving task-relevant signals. DAB-GNN Lee et al. (2025) further disentangles attribute bias, structural bias, and their interactions, explicitly amplifies these components, and then performs distribution alignment and contrastive regularization for debiasing, achieving fine-grained fairness control in an end-to-end manner.

Overall, pre-processing methods mitigate unfairness at the data level by modifying distributions or structures before training, while in-processing methods act directly within the learning process through fairness-aware regularization, loss constraints, or architectural redesigns. Together, these strategies highlight complementary perspectives on mitigating bias in GNNs.

### A.2 FAIRNESS IN INFORMATION BOTTLENECK

The fundamental principle of the IB framework is to identify a minimal sufficient representation that optimizes the compression of input data while preserving only the most critical information necessary for subsequent tasks Kawaguchi et al. (2023). The application of this principle to graph-structured data poses unique challenges, as the non-independent and identically distributed (NIID) nature of graphs complicates traditional optimization methods Xie et al. (2024). In order to address this issue, researchers have proposed the Graph Information Bottleneck (GIB) model Wu et al. (2020a), which extends IB to graph learning by simultaneously compressing node features and structural information.

In the pursuit of fair graph representation learning, IB theory demonstrates considerable potential due to its capacity to accurately quantify and regulate the information contained within a representation, thereby facilitating a more optimal balance between model utility and fairness Jiang et al. (2024). In order to achieve this objective in a stable manner, frameworks such as GRAFair Zhang et al. (2025) employ a variational graph autoencoder architecture. This architecture renders the optimization process tractable and effectively circumvents the instability issues that are prevalent in adversarial learning. Recent research has combined the IB principle with disentanglement learning and counterfactual augmentation to enhance the debiasing process. For instance, FDGIB Zheng et al. (2024) employs IB theory to direct the model in decomposing node representations into two distinct subspaces: one correlated with the sensitive attribute and one independent of it. Despite these

advances, single-view or single-embedding processing remains limited in its capacity. Graph bias is multi-source, and reliance on a single desensitized representation can lead to the conflation of signals from different origins, resulting in under-correction and residual leakage.

## B PRELIMINARIES

### B.1 FAIRNESS METRICS

To evaluate the fairness of our model, we focus on Group Fairness, which aims to ensure that the model's predictions are not biased against any specific group. In the node classification task, we adopt two widely used fairness metrics: Demographic Parity Dwork et al. (2012) and Equal Opportunity Hardt et al. (2016).

Here, we consider a common binary classification scenario where $s \in \{0, 1\}$ represents the sensitive attribute of a node (e.g., two different demographic groups), $y \in \{0, 1\}$ denotes the ground-truth label, and $\hat{y} \in \{0, 1\}$ is the predicted label given by the model.

#### B.1.1 DEMOGRAPHIC PARITY (DP)

The core idea of Demographic Parity is that the model prediction $\hat{y}$, should be statistically independent of the sensitive attribute $s$. This principle asserts that the probability of receiving a positive outcome should be the same for all demographic groups, regardless of their true label. This principle is formally expressed as:

$$P(\hat{y} = 1 | s = 0) = P(\hat{y} = 1 | s = 1) \tag{13}$$

In practice, we measure the violation of this metric by calculating the absolute difference in positive prediction rates between groups, known as the DP Difference ($\Delta_{DP}$). A smaller value indicates a fairer model.

$$\Delta_{DP} = |P(\hat{y} = 1 | s = 0) - P(\hat{y} = 1 | s = 1)| \tag{14}$$

#### B.1.2 EQUAL OPPORTUNITY (EO)

Equal Opportunity imposes a more targeted requirement: for nodes that genuinely belong to the positive class ($y = 1$), the model's prediction $\hat{y}$ should be conditionally independent of the sensitive attribute $s$. In other words, this ensures that individuals who are truly positive have an equal chance of being correctly identified, regardless of their group membership.

This is equivalent to requiring that the True Positive Rate (TPR) be consistent across different groups, which is formally defined as:

$$P(\hat{y} = 1 | s = 0, y = 1) = P(\hat{y} = 1 | s = 1, y = 1) \tag{15}$$

Similarly, we quantify the violation of this metric by calculating the absolute difference in the True Positive Rates between groups, referred to as the EO Difference ($\Delta_{EO}$). A value closer to zero signifies better performance in terms of equal opportunity.

$$\Delta_{EO} = |P(\hat{y} = 1 \mid s = 0, y = 1) - P(\hat{y} = 1 \mid s = 1, y = 1)| \tag{16}$$

### B.2 Multi-view Information Bottleneck

In case of dealing with complex information systems like graph data $\mathcal{G}$, a single source of information is often insufficient to capture the full spectrum of factors required for decision-making. The predicted label of a node is typically influenced by multiple information sources, or views, such as its intrinsic attributes $\mathbf{X}$, topological structure $\mathbf{A}$, and even global information diffusion patterns. The traditional single-view IB Kawaguchi et al. (2023)theory provides a core principle for understanding the trade-off between accuracy and compression. Its objective is to derive an optimal representation $\mathbf{Z}$, by maximizing the mutual information between the target labels $Y$ and the representation $\mathbf{Z}$, while simultaneously minimizing the mutual information between an input (e.g., $\mathbf{X}$) and the representation $\mathbf{Z}$. However, when information originates from multiple heterogeneous views $\{\mathcal{G}_1, \mathcal{G}_2, \ldots, \mathcal{G}_V\}$, a more powerful theoretical tool is needed to guide the learning process.

To this end, we introduce and extend the Multi-view Information Bottleneck (MIB) Chaudhuri et al. (2009) principle. The core idea of MIB is to learn a fused and compact representation matrix $\mathbf{Z}$, from multiple information views. This representation must satisfy two primary objectives:

- **Maximize Compression:** The representation $\mathbf{Z}$ must maximally compress the total information from all views to filter out task-irrelevant redundancy and noise.
- **Maximize Relevance:** Simultaneously, $\mathbf{Z}$ must preserve the most sufficient information relevant to the downstream prediction task (represented by the labels $Y$) to ensure the model's predictive performance.

## C  Methodology

After constructing the three disentangled views, our objective is to learn a fair and compressed representation. To this end, we adapt and extend the principles of the CFB Gálvez et al. (2021). The core objective is to learn a mapping from a graph view $\mathcal{G}_{\text{view}}$ to a latent representation $\mathbf{Z}_{\text{view}}$. This mapping aims to minimize the information from $\mathcal{G}_{\text{view}}$ contained in $\mathbf{Z}_{\text{view}}$ while maximizing the task-relevant information for $Y$ that is independent of the sensitive attribute $\mathbf{S}$.

For our Multi-view model, the total optimization objective can be written in the following Lagrangian form:

$$\min_{P(\mathbf{Z}|\{\mathcal{G}_{\text{view}}\})} \{I(\mathbf{S}; \mathbf{Z}) + I(\{\mathcal{G}_{\text{view}}\}; \mathbf{Z}|\mathbf{S}, Y) - \beta I(Y; \mathbf{Z}|\mathbf{S})\} \tag{17}$$

where $\mathbf{Z}$ is the final representation fused from the three view-specific representations: $\mathbf{Z}_{\text{feat}}$, $\mathbf{Z}_{\text{struct}}$, and $\mathbf{Z}_{\text{diff}}$. Based on information-theoretic properties and the Markov chain assumption $(\mathbf{S}, Y) \leftrightarrow \{\mathcal{G}_{\text{view}}\} \to \mathbf{Z}$, this objective can be simplified to:

$$\min_{P(\mathbf{Z}|\{\mathcal{G}_{\text{view}}\})} \{I(\{\mathcal{G}_{\text{view}}\}; \mathbf{Z}) - \gamma I(Y; \mathbf{Z}|\mathbf{S})\}, \quad \text{where } \gamma = \beta + 1 \tag{18}$$

This formulation intuitively expresses our dual objectives: (1) Compression: minimizing the total information $I(\{\mathcal{G}_{\text{view}}\}; \mathbf{Z})$ extracted from all views; and (2) Fair

Prediction: maximizing the task-relevant information $I(Y; \mathbf{Z}|\mathbf{S})$ contained in the representation $\mathbf{Z}$, conditioned on the sensitive attribute $\mathbf{S}$.

As illustrated in the overall framework, each view $\mathcal{G}_{\text{view}}$ is processed by an independent variational graph encoder $g_{\theta_{\text{view}}}$ Kipf & Welling (2016). The encoder outputs the parameters of the latent distribution for each node, namely the mean vector $\boldsymbol{\mu}_{\text{view}}$ and the log-variance vector $\log \boldsymbol{\sigma}_{\text{view}}$. We utilize the reparameterization trick to sample from this distribution, which ensures that gradients can be backpropagated through the sampling process:

$$\mathbf{Z}_{\text{view}} = \boldsymbol{\mu}_{\text{view}} + \boldsymbol{\sigma}_{\text{view}} \odot \boldsymbol{\epsilon}, \quad \text{where } \boldsymbol{\epsilon} \sim \mathcal{N}(0, \mathbf{I}) \tag{19}$$

After obtaining the latent representations for the three views, we perform an initial fusion via element-wise addition. The result is then passed through a projector layer, implemented as a Multi-Layer Perceptron (MLP), to learn more complex interactions and to generate the final unified representation, $\mathbf{Z}_{\text{proj}}$:

$$\mathbf{Z}_{\text{proj}} = \text{Projector}(\mathbf{Z}_{\text{feat}}, \mathbf{Z}_{\text{struct}}, \mathbf{Z}_{\text{diff}}) \tag{20}$$

### C.1 BALANCE DIFFSION VIEW

Diffusion Views help us to identify potential dynamic deviations that may occur as information propagates across a graph. To prevent sensitive attributes from becoming biased during diffusion, we have implemented proactive intervention measures to balance this bias, as shown in the Figure 5.

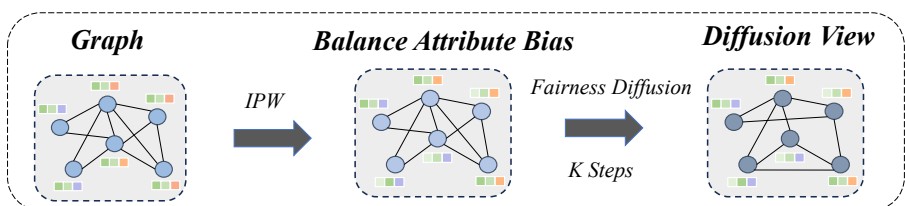

Figure 5: The generation process for the Diffusion View begins with the original attributed graph. First, attribute bias is balanced using Inverse Propensity Weighting (IPW). A K-step fairness diffusion process is then executed on this basis to ultimately generate the Diffusion View, which incorporates fair neighborhood information.

## D EXPERIMENTS

### D.1 BASELINE

The methods under discussion can be categorized as follows: FairGNN Dai & Wang (2021) and FairVGNN Wang et al. (2022) belong to the class of adversarial representation learning methods; NIFTY Agarwal et al. (2021), EDITS Dong et al. (2022), and FairGB Li et al. (2024b) are data augmentation-based methods; GRAFair Zhang et al. (2025) is an information bottleneck-based method; and DAB-GNN Lee et al. (2025) is a disentangled representation learning approach. For specific details, please refer to the appendix. The following is a detailed introduction to all baseline methods.

- **FairGNN:** FairGNN Dai & Wang (2021) is a framework designed to eliminate discrimination in GNNs by using adversarial learning and a sensitive attribute estimator to achieve fair node classification even with limited sensitive attribute information.

- **NIFTY:** NIFTY Agarwal et al. (2021) is a unified approach that promotes fairness and stability in GNN representations by leveraging counterfactual perturbations and layer-wise weight normalization to ensure robust and unbiased graph embeddings.

- **EDITS:** EDITS Dong et al. (2022) mitigates bias in attributed networks for GNNs by optimizing attribute re-weighting and structural adjustments to reduce disparities between demographic groups while preserving downstream task performance.

- **FairVGNN:** FairVGNN Wang et al. (2022) is a framework that integrates adversarial learning with weight clipping to mitigate sensitive attribute leakage.

- **FairGB:** FairGB Li et al. (2024b) addresses unfairness in GNNs through group re-balancing techniques, such as counterfactual node mixup and contribution alignment, to ensure balanced influence from different demographic groups during training.

- **GRAFair:** GRAFair Zhang et al. (2025) is a variational graph auto-encoder-based framework that achieves stable fairness by minimizing sensitive information in representations via a conditional fairness bottleneck, balancing utility and debiasing without adversarial methods.

- **DAB-GNN:** DAB-GNN Lee et al. (2025) promotes fair GNN representations by disentangling and amplifying attribute, structure, and potential biases, then debiasing them to minimize subgroup distribution differences.

### D.2 ABLATION STUDY

we conducted a series of ablation studies. Our proposed framework is fundamentally an implementation of the Multi-view conditional information bottleneck, which aims to maximize task-relevant fair information while minimizing irrelevant information from the Multi-view inputs. To systematically evaluate the contribution of each core component, we constructed three key variants: FairMIB w/o m (without information compression), FairMIB w/o s (without the conditional constraint of the bottleneck), and FairMIB w/o c (without the Multi-view consistency constraint).

First, we validate the role of the conditional information bottleneck's core mechanism by removing it (FairMIB w/o s). The objective of this component is to maximize the information in the representation that is relevant to the task $\mathbf{Y}$ but independent of the sensitive attribute $\mathbf{S}$, i.e., $I(\mathbf{Y}; \mathbf{Z}|\mathbf{S})$. As shown in Table 3, removing this module leads to a significant decline in model fairness. Across all datasets, the fairness metrics of FairMIB w/o s worsened significantly; for instance, on the Bail dataset, its DP metric worsening by over 30%. This indicates that conditioning the decoder on the sensitive attribute during training is crucial for compelling the

encoder to learn a truly fair representation, as it effectively weakens the model's ability to capture and rely on sensitive information, thereby ensuring the achievement of the fairness objective.

Second, we investigate the contribution of information compression using the Fair-MIB w/o m variant. This module corresponds to the objective of minimizing mutual information between input views and the representation, $I(\{\mathcal{G}_{\text{view}}\}; \mathbf{Z})$, and is designed to prevent the model from learning redundant or harmful biased information. The results show that removing this module leads to a substantial decline in both predictive performance and fairness. This phenomenon was particularly pronounced in the Pokec-z dataset, where the DP metric increased from being nearly 40%, and the utility also decreased. This demonstrates that the compression of redundant information effectively improves both utility and fairness by forcing the model to learn a compact representation, thereby filtering out bias-propagating information from the input views.

Finally, we assess the role of the Multi-view consistency constraint by evaluating the FairMIB w/o c variant. The removal of this constraint leads to a noticeable decline in fairness performance across datasets, with the effect being particularly severe on the Pokec-n dataset, where both DP and EO metrics deteriorate beyond those observed in other ablation variants (Table 3). These results underscore the importance of enforcing semantic alignment between representations of different views through contrastive learning. By aligning the latent spaces across views, the model is guided to learn a robust and coherent shared representation, preventing individual view encoders from independently capturing conflicting or biased patterns. This alignment is critical for the overall debiasing process, ensuring that the learned representations are both fair and consistent.

The above ablation study results validate the effectiveness of the three core components of the FairMIB model: the conditional information bottleneck, information compression, and the Multi-view consistency constraint. These components work synergistically through their respective mechanisms, collectively mitigating sensitive attribute bias by ensuring fairness objectives, filtering out irrelevant information, and aligning Multi-view representations.

### D.3 MULTI-VIEW ANALYSIS

To address RQ3, we conducted a series of ablation experiments to assess the contribution of each view by selectively removing the Diffusion View (w/o Diffusion View), the Feature View (w/o Feature View), or the Structural View (w/o Structural View). The results presented in Table 4 indicate that the three views provide complementary information, and their joint utilization is critical for achieving an optimal balance between utility and fairness. Removing any single view typically results in a significant decline in model performance. For example, on the Bail dataset, excluding the Feature View leads to a drop of over 10% in F1-score, while fairness metrics DP and EO also deteriorate sharply, highlighting the essential role of original node attributes in maintaining both baseline predictive performance and fair decision-making.

Table 3: Results of FairMIB ablations on five datasets

| Datasets | Method | Acc (↑) | F1-score (↑) | AUC (↑) | ΔDP (↓) | ΔEO (↓) |
|---|---|---|---|---|---|---|
| german | FairMIB w/o m | $70.00 \pm 0.00$ | $82.35 \pm 0.00$ | $\underline{60.62 \pm 4.78}$ | $0.51 \pm 0.02$ | $\underline{0.44 \pm 0.02}$ |
| | FairMIB w/o s | $70.00 \pm 0.00$ | $82.35 \pm 0.00$ | $55.85 \pm 2.60$ | $0.87 \pm 0.22$ | $0.63 \pm 0.11$ |
| | FairMIB w/o c | $\underline{70.00 \pm 0.00}$ | $\underline{82.35 \pm 0.00}$ | $59.25 \pm 4.25$ | $\underline{0.42 \pm 0.12}$ | $0.55 \pm 0.01$ |
| | FairMIB | $\mathbf{70.24 \pm 0.48}$ | $\mathbf{82.45 \pm 0.20}$ | $\mathbf{65.55 \pm 1.61}$ | $\mathbf{0.38 \pm 0.76}$ | $\mathbf{0.17 \pm 0.34}$ |
| bail | FairMIB w/o m | $\underline{84.32 \pm 1.34}$ | $77.65 \pm 1.94$ | $87.76 \pm 1.55$ | $\mathbf{0.96 \pm 0.58}$ | $1.83 \pm 1.44$ |
| | FairMIB w/o s | $84.12 \pm 3.20$ | $77.97 \pm 4.11$ | $87.93 \pm 3.16$ | $1.60 \pm 0.36$ | $\underline{1.52 \pm 0.81}$ |
| | FairMIB w/o c | $84.21 \pm 3.05$ | $\underline{78.11 \pm 3.83}$ | $\underline{88.08 \pm 2.47}$ | $1.34 \pm 1.25$ | $1.55 \pm 0.42$ |
| | FairMIB | $\mathbf{85.62 \pm 0.81}$ | $\mathbf{80.10 \pm 1.25}$ | $\mathbf{89.18 \pm 2.15}$ | $\underline{1.23 \pm 0.49}$ | $\mathbf{1.17 \pm 0.45}$ |
| credit | FairMIB w/o m | $78.04 \pm 0.63$ | $87.51 \pm 0.10$ | $71.04 \pm 1.03$ | $1.27 \pm 1.70$ | $0.74 \pm 1.00$ |
| | FairMIB w/o s | $77.92 \pm 0.06$ | $87.57 \pm 0.02$ | $71.34 \pm 0.90$ | $0.70 \pm 0.33$ | $0.66 \pm 0.21$ |
| | FairMIB w/o c | $\underline{78.07 \pm 0.29}$ | $\underline{87.59 \pm 0.14}$ | $\underline{71.62 \pm 0.97}$ | $\underline{0.62 \pm 0.55}$ | $\underline{0.28 \pm 0.31}$ |
| | FairMIB | $\mathbf{78.57 \pm 0.86}$ | $\mathbf{87.79 \pm 0.27}$ | $\mathbf{73.21 \pm 0.51}$ | $\mathbf{0.40 \pm 0.69}$ | $\mathbf{0.24 \pm 0.48}$ |
| Pokec-z | FairMIB w/o m | $65.70 \pm 1.85$ | $67.54 \pm 0.84$ | $73.11 \pm 0.89$ | $3.41 \pm 2.43$ | $2.90 \pm 1.42$ |
| | FairMIB w/o s | $64.54 \pm 2.00$ | $68.24 \pm 0.90$ | $72.10 \pm 1.24$ | $2.56 \pm 1.44$ | $\underline{1.38 \pm 0.96}$ |
| | FairMIB w/o c | $\mathbf{66.56 \pm 1.94}$ | $\mathbf{69.04 \pm 3.45}$ | $\mathbf{74.68 \pm 1.81}$ | $\underline{1.43 \pm 0.67}$ | $2.14 \pm 1.57$ |
| | FairMIB | $\underline{66.16 \pm 1.43}$ | $\underline{68.86 \pm 1.40}$ | $\underline{73.15 \pm 1.64}$ | $\mathbf{0.69 \pm 0.26}$ | $\mathbf{0.52 \pm 0.38}$ |
| Pokec-n | FairMIB w/o m | $65.40 \pm 1.72$ | $67.33 \pm 1.58$ | $72.72 \pm 2.04$ | $2.04 \pm 1.17$ | $\underline{1.90 \pm 1.26}$ |
| | FairMIB w/o s | $\mathbf{66.96 \pm 2.25}$ | $62.75 \pm 2.73$ | $\underline{73.36 \pm 1.94}$ | $\underline{1.98 \pm 0.41}$ | $2.29 \pm 1.16$ |
| | FairMIB w/o c | $\underline{66.48 \pm 5.14}$ | $\underline{67.58 \pm 1.07}$ | $\mathbf{74.21 \pm 3.80}$ | $2.73 \pm 1.22$ | $2.80 \pm 1.74$ |
| | FairMIB | $66.22 \pm 1.09$ | $\mathbf{69.02 \pm 1.91}$ | $73.28 \pm 1.02$ | $\mathbf{1.12 \pm 0.76}$ | $\mathbf{0.92 \pm 0.90}$ |

Interestingly, the relative importance of the Structural and Diffusion Views varies across datasets. On the Pokec-z and Pokec-n social network datasets, which have authentic topological structures, removing the Structural View results in catastrophic performance degradation, with DP increasing by over 380% and 205%, respectively. This indicates that, in these topologies, the primary source of bias originates from the graph structure itself, making it crucial to model and debias structural information directly. Conversely, in the Credit dataset, removing the Diffusion View has the most pronounced negative effect on fairness, with DP rising by 205%. This suggests that, in this context, bias predominantly propagates through multi-hop connections, underscoring the critical role of our designed Diffusion View in capturing and correcting such biases.

Overall, these findings demonstrate that no single view universally dominates across all datasets, as the sources of bias differ depending on the graph type. This further validates the necessity and effectiveness of our Multi-view decoupling framework.

### D.4 HYPER-PARAMETER SENSITIVITY ANALYSIS

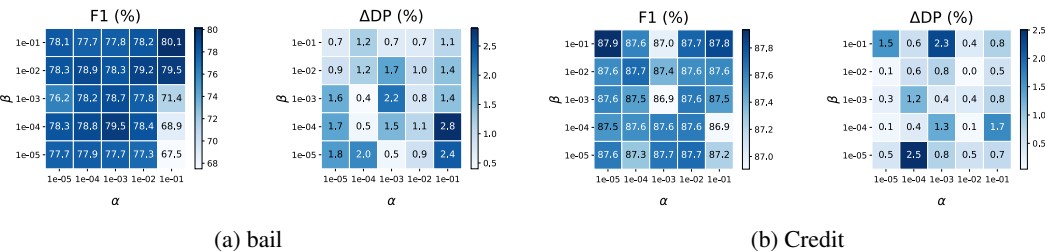

(a) bail  (b) Credit

Figure 6: Parameter sensitivity results on two datasets.

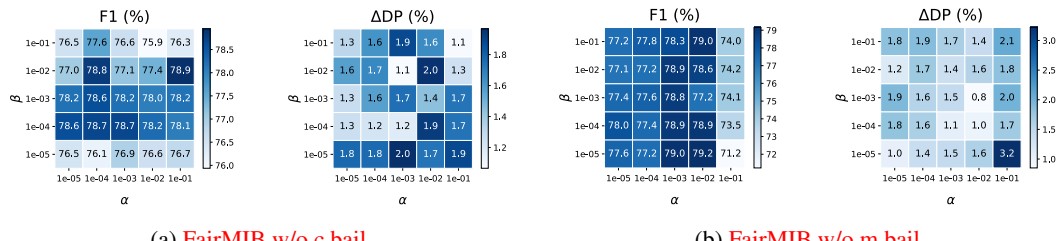

(a) FairMIB w/o c bail          (b) FairMIB w/o m bail

Figure 7: Parameter sensitivity results on two variants.

Table 4: Ablation over multiple views.

| Datasets | Method | Acc (↑) | F1-score (↑) | AUC (↑) | ΔDP (↓) | ΔEO (↓) |
|---|---|---|---|---|---|---|
| german | w/o Diffusion View | $70.16 \pm 0.20$ | $82.35 \pm 0.03$ | $57.31 \pm 4.01$ | $0.51 \pm 0.92$ | $0.36 \pm 0.66$ |
| | w/o Feature View | $70.00 \pm 0.00$ | $82.35 \pm 0.00$ | $53.40 \pm 3.41$ | $0.50 \pm 0.11$ | $0.20 \pm 0.01$ |
| | w/o Structure View | $70.00 \pm 0.00$ | $82.35 \pm 0.00$ | $63.07 \pm 3.69$ | $\mathbf{0.03 \pm 0.01}$ | $\mathbf{0.02 \pm 0.03}$ |
| | FairMIB | $\mathbf{70.24 \pm 0.48}$ | $\mathbf{82.45 \pm 0.20}$ | $\mathbf{65.55 \pm 1.61}$ | $0.38 \pm 0.76$ | $0.17 \pm 0.34$ |
| bail | w/o Diffusion View | $\mathbf{86.19 \pm 2.53}$ | $80.69 \pm 3.30$ | $\mathbf{89.44 \pm 1.85}$ | $1.12 \pm 0.68$ | $3.22 \pm 3.14$ |
| | w/o Feature View | $83.21 \pm 5.40$ | $72.53 \pm 14.73$ | $87.13 \pm 2.98$ | $2.61 \pm 1.74$ | $2.38 \pm 1.81$ |
| | w/o Structure View | $83.76 \pm 1.49$ | $\mathbf{86.79 \pm 1.65}$ | $86.50 \pm 1.30$ | $\mathbf{0.79 \pm 1.18}$ | $\mathbf{0.93 \pm 1.19}$ |
| | FairMIB | $85.62 \pm 0.81$ | $80.10 \pm 1.25$ | $89.18 \pm 2.15$ | $1.23 \pm 0.49$ | $1.17 \pm 0.45$ |
| credit | w/o Diffusion View | $78.11 \pm 0.48$ | $87.53 \pm 0.06$ | $69.29 \pm 2.20$ | $1.22 \pm 2.39$ | $0.73 \pm 1.44$ |
| | w/o Feature View | $78.46 \pm 0.78$ | $87.70 \pm 0.28$ | $71.84 \pm 0.95$ | $0.55 \pm 0.34$ | $0.56 \pm 0.36$ |
| | w/o Structure View | $\mathbf{78.94 \pm 0.88}$ | $\mathbf{87.87 \pm 0.24}$ | $72.19 \pm 0.93$ | $1.17 \pm 0.56$ | $0.70 \pm 0.44$ |
| | FairMIB | $78.57 \pm 0.86$ | $87.79 \pm 0.27$ | $\mathbf{73.21 \pm 0.51}$ | $\mathbf{0.40 \pm 0.69}$ | $\mathbf{0.24 \pm 0.48}$ |
| Pokec-z | w/o Diffusion View | $62.19 \pm 4.16$ | $69.03 \pm 1.79$ | $71.90 \pm 1.59$ | $1.19 \pm 0.96$ | $1.62 \pm 1.53$ |
| | w/o Feature View | $\mathbf{67.57 \pm 1.86}$ | $67.48 \pm 2.90$ | $73.03 \pm 1.43$ | $1.37 \pm 0.79$ | $2.34 \pm 2.00$ |
| | w/o Structure View | $65.74 \pm 3.37$ | $\mathbf{70.78 \pm 1.15}$ | $\mathbf{75.17 \pm 3.02}$ | $3.33 \pm 1.09$ | $1.24 \pm 0.65$ |
| | FairMIB | $66.16 \pm 1.43$ | $68.86 \pm 1.40$ | $73.15 \pm 1.64$ | $\mathbf{0.69 \pm 0.26}$ | $\mathbf{0.52 \pm 0.38}$ |
| Pokec-n | w/o Diffusion View | $65.51 \pm 0.70$ | $66.05 \pm 0.62$ | $71.68 \pm 0.85$ | $1.29 \pm 0.89$ | $1.54 \pm 1.48$ |
| | w/o Feature View | $\mathbf{68.58 \pm 0.85}$ | $67.96 \pm 3.21$ | $71.36 \pm 1.30$ | $1.61 \pm 1.22$ | $2.18 \pm 1.69$ |
| | w/o Structure View | $67.15 \pm 2.42$ | $66.39 \pm 0.77$ | $\mathbf{73.83 \pm 2.84}$ | $3.42 \pm 2.78$ | $2.75 \pm 2.57$ |
| | FairMIB | $66.22 \pm 1.09$ | $\mathbf{69.02 \pm 1.91}$ | $73.28 \pm 1.02$ | $\mathbf{1.12 \pm 0.76}$ | $\mathbf{0.92 \pm 0.90}$ |

We conduct a sensitivity analysis of FairMIB with respect to two hyperparameters, $\alpha$ and $\beta$. In FairMIB, these hyperparameters regulate the relative contributions of information compression and view alignment. Specifically, we vary the values of $\alpha$ and $\beta$ within $\{10^{-1}, 10^{-2}, 10^{-3}, 10^{-4}, 10^{-5}\}$ on the bail, Credit, Pokec-z, and Pokec-n datasets. The results of this analysis are presented in Figure 6. Overall, the performance of FairMIB remains relatively stable across a broad range of $\alpha$ and $\beta$. Nevertheless, when $\alpha$ and $\beta$ are set to excessively large values, performance degradation may occur due to over-compression of information and overly strict view alignment. These findings highlight the necessity of balancing the two components and suggest that selecting $\alpha$ and $\beta$ from the range of $10^{-3}$ to $10^{-5}$ offers a preferable trade-off between utility and fairness. In Figure 7, we performed a hyperparameter analysis on the two variants for bail to ensure that the hyperparameters do not affect the ablation experiments.

# E  USE OF LLMS

Yes, to aid or polish writing. Details are described in the paper.

