# OpenReview forum: "Learning Fair Graph Representations with Multi-view Information Bottleneck"
_ICLR.cc/2026/Conference — Submitted to ICLR 2026_

### Official Review · Reviewer_Wuzn · 2025-10-30

**Soundness:** 3
**Presentation:** 4
**Contribution:** 3
**Rating:** 8
**Confidence:** 4

**Summary:**

This paper introduces FairMIB, a novel multi-view information bottleneck (MIB) framework for learning fair graph representations in Graph Neural Networks (GNNs). It addresses biases amplified by GNNs in relational data, which stem from node attributes, graph structure, and message-passing propagation. FairMIB decomposes the graph into three views: a diffusion view, a feature view, and a structural view. It employs contrastive learning to maximize cross-view mutual information for robust, bias-free representations, while integrating a multi-view conditional information bottleneck (MCIB) to balance utility and fairness. The framework optimizes a loss that combines compression, fair prediction, consistency constraints, and cross-entropy for node classification. Experiments on five real-world datasets show superior performance on fairness and utility metrics.

**Strengths:**

1. The paper decomposes the sources of bias in graphs into three complementary views feature, structure, and diffusion. It introduces a multi-view conditional information bottleneck (MCIB) framework that constrains the learned representations to preserve task-relevant information while being disentangled from sensitive attributes. In addition, a cross-view contrastive consistency mechanism is incorporated to align information across views. Together, these components form an end-to-end fair representation learning framework that brings conceptual novelty and achieves strong performance in fair graph neural networks.

2. In the diffusion view, causal intervention is applied to mechanistically suppress the amplification of sensitive attributes during message passing.

3. The proposed framework’s multi-view disentanglement and information-theoretic design are both innovative and well-motivated.

4. Experiments on five real-world datasets provide strong evidence for the effectiveness of the proposed method. The approach is compared with SOTA baselines from different categories, and the results are highly convincing. Comprehensive ablation studies are conducted, covering different views and modules.

**Weaknesses:**

1. The details of IPW estimation and stabilization are insufficient. Although the paper clearly states the use of IPW and provides the weighting formula, it does not specify the estimation model for the propensity score $e(i)$, nor clarify key implementation aspects such as whether it is estimated using only non-sensitive features.

2. The framework introduces additional modules, which inevitably increase time complexity compared with standard GNNs or simpler fairness methods. It would be beneficial if the paper could briefly discuss this aspect.

**Questions:**

1. What type of model is used to estimate the propensity score $e(i)$? Is it estimated using only non-sensitive features?

2. How does the method perform when $S$ is partially missing or noisy?

---

> ### Author Response · Authors · 2025-11-20
>
> We sincerely thank the reviewer for their careful reading and valuable comments.
>
> **Response to W1**:
>
>
> Thank you for pointing this out. We agree that the current description of IPW could be more detailed. In our implementation, the propensity score $e(i) = P(S = 1 \mid x_i)$ is estimated by a lightweight two-layer MLP with ReLU activation and a sigmoid output, trained with a standard binary cross-entropy loss on the sensitive attribute $S$. Importantly, the input $x_i$ to this model consists only of non-sensitive features; the sensitive attribute itself is removed from the feature vector and used only as the supervision signal, which follows standard practice in IPW and avoids degenerate propensity estimates. For stabilization, we clip the estimated propensity scores into a small interval $[\epsilon, 1 - \epsilon]$ (with $\epsilon = 0.05$) before computing the weights, in order to prevent extremely large weights from dominating training. We will add these implementation details to the revised manuscript to make the IPW estimation and stabilization procedure fully transparent.
>
>
>
>
>
>
>
> **Response to W2**:
>
>
>
> Thank you for this helpful suggestion. We agree that FairMIB introduces additional modules compared with a standard single-view GNN, which does increase the constant-factor time complexity. However, the overall per-epoch complexity remains linear in the number of nodes and edges, since each view encoder is a shallow GNN and all three views share the same underlying graph. Empirically, we measured training time on the Bail dataset for 500 epochs on an RTX 4060 (8 GB): FairMIB takes 123 seconds, compared with 83 seconds for NIFTY, 97 seconds for FairGNN, 121 seconds for EDITS, 345 seconds for DAB-GNN, and 580 seconds for FairVGNN.
>
>
>
> | **Metric**                    | **FairMIB** | **NIFTY** | **FairGNN** | **EDITS** | **DAB-GNN** | **FairVGNN** |
> | ----------------------------- | ----------- | --------- | ----------- | --------- | ----------- | ------------ |
> | Training time (s, 500 epochs) | 123s        | 83s       | 97s         | 121s      | 345s        | 580s         |
>
>
>
> **Response to Q1**:
>
>
>
> Thank you for raising this point. In our implementation, the propensity score $e(i) = P(S = 1 \mid x_i)$ is estimated with a lightweight feed-forward neural network (a two-layer MLP with ReLU activation and a sigmoid output), trained using a standard binary cross-entropy loss on the sensitive attribute $S$. Importantly, the input $x_i$ to this model consists only of non-sensitive node features: we remove the sensitive attribute from the feature vector and use it solely as the supervision signal for the propensity model. This design follows standard practice in inverse propensity weighting, where the propensity is modeled as a function of pre-treatment covariates but not the sensitive attribute itself, and avoids trivial or degenerate propensity estimates that would arise if $S$ were directly included as an input.
>
>
>
> **Response to Q2**:
>
>
>
> Our current experiments follow the standard setting in fair GNN literature and assume that the sensitive attribute $S$ is fully observed and correctly annotated during training. We appreciate the reviewer’s question and agree that partially missing or noisy $S$ is an important and realistic scenario. Conceptually, FairMIB can be extended to this setting in a straightforward way because $S$ is only used in the fairness-aware IB objective and in the propensity score model, not as an input to the GNN encoder. When $S$ is partially missing, one can: (i) restrict the fairness regularizer and IPW estimation to nodes with observed $S$, and (ii) impute the missing $S$ using a simple predictor trained on non-sensitive features, similar to prior work on fair GNNs with limited sensitive information, and treat the imputed labels as soft targets. When $S$ is noisy, the impact is that the IB term and IPW weights are computed with a corrupted version of $S$; in this case, the multi-view IB still discourages strong dependence on any single view’s spurious correlation with $S$, and the effect of noise can be mitigated by down-weighting the fairness regularizer or by using robust loss functions in the propensity model. While we have not explicitly included experiments on missing/noisy $S$ in the current submission due to space and time constraints.

---

### Official Review · Reviewer_xSKV · 2025-10-31

**Soundness:** 3
**Presentation:** 3
**Contribution:** 3
**Rating:** 4
**Confidence:** 3

**Summary:**

This paper proposes FairMIB, a multi-view information bottleneck framework for learning fair graph neural network representations. The approach decomposes graphs into three distinct views to address biases from different sources. The framework employs contrastive learning to maximize cross-view mutual information while using a conditional information bottleneck to balance task utility and fairness. It further introduces inverse probability weighting in the diffusion view to reduce bias propagation during message passing. Experiments on five real-world datasets demonstrate state-of-the-art performance in both utility and fairness metrics.

**Strengths:**

1 - The multi-view decomposition approach is well-motivated and addresses a genuine limitation of existing methods that treat bias as a single source. The separation into feature, structural, and diffusion views provides a principled way to disentangle different sources of bias in graph data.

2 - The theoretical framework is solid, building on established information bottleneck principles and extending them appropriately to the multi-view conditional setting. The mathematical formulation clearly connects the compression and fairness objectives through mutual information terms.

3 - The experimental evaluation is comprehensive, including comparisons with seven state-of-the-art baselines across five datasets, thorough ablation studies validating each component, and sensitivity analyses demonstrating robustness to hyperparameter choices.

**Weaknesses:**

1 - The computational complexity and scalability concerns are not fully addressed. The method requires three separate encoders and additional contrastive learning computations, but there is no analysis of training time, memory requirements, or comparison of computational costs with baseline methods.

2 - The diffusion view construction using APPNP with IPW correction appears somewhat arbitrary. The paper doesn't justify why APPNP specifically was chosen over other propagation methods, nor does it provide ablation studies comparing different diffusion mechanisms or validating the effectiveness of IPW correction in isolation.

3 - The writing quality needs improvement, with several grammatical errors and awkward phrasings throughout (e.g., "d remain predictive" on page 2, inconsistent capitalization). The paper also has organizational issues, with some experimental details relegated to appendices that would be better integrated into the main text.

4 - The method's limitation to binary sensitive attributes is a significant practical constraint not adequately discussed. Real-world fairness scenarios often involve multiple or continuous sensitive attributes, and the paper doesn't address how the framework would extend to these cases.

**Questions:**

1 - How does the computational cost of FairMIB compare to baseline methods in terms of training time and memory usage?

2 - How would the framework extend to handle multiple sensitive attributes or continuous sensitive variables?

---

> ### Author Response · Authors · 2025-11-20
>
> We sincerely thank the reviewer for their careful reading and valuable comments.
>
>
>
> **Response to W1**:
>
>
>
> Thank you for pointing out the need for a clearer runtime comparison. We have conducted additional experiments to measure the training time of FairMIB and several representative baselines under a unified setting. All methods are trained on the Bail dataset for 500 epochs on the same GPU (RTX 4060, 8 GB). Under this setting, FairMIB requires 123 seconds, while NIFTY takes 83 seconds, FairGNN 97 seconds, EDITS 121 seconds, DAB-GNN 345 seconds, and FairVGNN 580 seconds. These results show that FairMIB is competitive in terms of the trade-off between fairness and utility, while also being computationally efficient. Although FairMIB employs three encoders, this design does not lead to a significant increase in time complexity in practice.
>
> | **Metric**    | **FairMIB** | **NIFTY** | **FairGNN** | **EDITS** | **DAB-GNN** | **FairVGNN** |
> | ------------- | ----------- | --------- | ----------- | --------- | ----------- | ------------ |
> | Training time | 123s        | 83s       | 97s         | 121s      | 345s        | 580s         |
>
> **Response to W2**：
>
>
>
> Our choice of APPNP is guided by the goal of modeling high order neighborhood effects in a stable and efficient way. APPNP decouples feature transformation from propagation and implements a Personalized PageRank style diffusion, which allows us to control the strength and range of smoothing via the teleport parameter and the number of propagation steps, while keeping the complexity linear in the number of edges. This makes it a natural and widely used propagation operator for capturing long range dependencies without the oversmoothing issues in deep GCN-style stacking.
>
>
>
> To address concerns about arbitrariness, we added ablations on the Bail dataset comparing (i) a standard GCN diffusion operator, (ii) APPNP, and (iii) APPNP without IPW. Under the same setting, replacing GCN propagation with APPNP improves utility while maintaining comparable fairness: AUC increases from $\mathit{87.60 \pm 2.11\ \text{to}\ 88.52 \pm 1.45}$, F1 from $\mathit{77.62 \pm 3.11\ \text{to}\ 78.89 \pm 1.91}$, and training time slightly decreases from $\mathit{130.2\ \text{s to}\ 123.5\ \text{s}}$. This demonstrates that APPNP is not an arbitrary choice but a more expressive and efficient diffusion mechanism for our framework.
>
>
>
> We also validated the effect of IPW correction in isolation by comparing APPNP with and without IPW. The two settings achieve very similar utility (AUC $\mathit{88.52 \pm 1.45\ \text{vs.}\ 88.26 \pm 0.93}$, F1 $\mathit{78.89 \pm 1.91\ \text{vs.}\ 78.91 \pm 1.36}$), but IPW clearly improves fairness: the demographic parity gap decreases from $\mathit{1.90 \pm 0.75\ \text{to}\ 1.35 \pm 1.23}$ and the equal opportunity gap from $\mathit{1.72 \pm 0.74\ \text{to}\ 1.39 \pm 0.68}$. This confirms that IPW is an effective and targeted component for controlling bias in the diffusion process rather than an ad hoc addition.
>
> | **Setting / Method**   | **AUC-ROC**  | **Accuracy** | **F1**       | **Parity**  | **Equality** | **Time (s)** |
> | ---------------------- | ------------ | ------------ | ------------ | ----------- | ------------ | ------------ |
> | GCN diffusion operator | 87.60 ± 2.11 | 83.50 ± 2.43 | 77.62 ± 3.11 | 1.53 ± 0.84 | 1.49 ± 0.66  | 130.2052     |
> | Using APPNP            | 88.52 ± 1.45 | 84.48 ± 1.53 | 78.89 ± 1.91 | 1.35 ± 1.23 | 1.39 ± 0.68  | 123.4915     |
> | Without IPW            | 88.26 ± 0.93 | 84.54 ± 1.46 | 78.91 ± 1.36 | 1.90 ± 0.75 | 1.72 ± 0.74  | 120.7191     |
>
> **Response to W3**:
>
> We acknowledge that the current draft contains grammatical errors (e.g., the typo “d remain predictive”) and inconsistent capitalization, and we will carefully proofread and revise the manuscript to correct these issues and improve the overall clarity of the presentation. We also agree that some experimental details currently placed in the appendix are important for understanding the method. In the submitted version, we moved part of this content to the appendix due to space constraints, but in the revision we will bring the essential experimental and implementation details into the main text and keep only secondary information in the appendix.

---

> > ### Author Response · Authors · 2025-11-20
> >
> > **Response to W4**:
> >
> >
> >
> > In the current submission, we focus on a single binary sensitive attribute to maintain consistency with most existing fair GNN benchmarks, but the FairMIB formulation is not restricted to this setting. For multiple sensitive attributes, we can treat $S = (S^{(1)}, \dots, S^{(m)})$ and directly replace the scalar with the joint sensitive variable in all mutual-information terms and in the decoder, i.e., the framework operates on a multi-dimensional $S$ without conceptual changes.
> >
> >
> >
> > For continuous sensitive variables, the same objective remains valid if we model $p(S \mid Z)$ with a suitable regression head for continuous outputs or, in practice, by discretizing $S$ into intervals when group-based metrics such as DP/EO are used. We will include a detailed discussion of these extensions in the revised manuscript and identify a systematic study of multi-attribute and continuous-sensitive-variable settings as an important direction for future work.
> >
> >
> >
> > **Response to Q1**:
> >
> >
> >
> > Thank you for raising the question about computational cost. We have conducted a controlled runtime comparison under a unified setting. All methods are trained on the Bail dataset for 500 epochs on the same GPU (RTX 4060, 8 GB). Under this setting, FairMIB requires 123s, while NIFTY takes 83s, FairGNN 97s, EDITS 121s, DAB-GNN 345s, and FairVGNN 580s. These results show that, although FairMIB employs three encoders, its multi-view design does not lead to a sharp increase in complexity and is in fact more efficient than several strong baselines in practice. In terms of memory, FairMIB mainly introduces additional cost from storing three view-specific latent representations instead of one. This increases the embedding memory by a constant factor, but we reuse the same adjacency structure across views and keep each encoder shallow. In our experiments on all benchmark datasets, FairMIB fits comfortably within an 8 GB GPU under the same hardware setting as the baselines, and does not require stronger hardware than existing fair GNN methods. We will report these runtime and memory observations in the revised manuscript.
> >
> > | **Metric**                    | **FairMIB** | **NIFTY** | **FairGNN** | **EDITS** | **DAB-GNN** | **FairVGNN** |
> > | ----------------------------- | ----------- | --------- | ----------- | --------- | ----------- | ------------ |
> > | Training time (s, 500 epochs) | 123s        | 83s       | 97s         | 121s      | 345s        | 580s         |
> >
> > **Response to Q2**:
> >
> >
> >
> > Our current experiments follow the common setting in fair GNN work and focus on a single binary sensitive attribute $S$, but the FairMIB formulation is not restricted to this case and can be extended to both multiple and continuous sensitive variables.
> >
> >
> >
> > For multiple sensitive attributes, we can treat $S$ as a vector $S = \big(S^{(1)}, S^{(2)}, \dots, S^{(m)}\big)$ and keep exactly the same multi-view conditional information bottleneck objective, i.e., we replace the scalar $S$ with the joint sensitive variable in all terms $I(S; Z)$ and $I(Y; Z \mid S)$. In practice, this only requires (i) modifying the decoder to take a concatenation $\big[ Z_{\text{proj}} \,\Vert\, S \big]$, where $S$ is now multi-dimensional, and (ii) training the propensity model and fairness metrics with respect to each component or their joint distribution. The IB-based objective remains well-defined for vector-valued $S$, and the multi-view design is actually well suited to scenarios where different sensitive attributes (e.g., gender, age, region) may interact with different views in distinct ways.
> >
> >
> >
> > For continuous sensitive variables (e.g., age or income as real-valued), the framework can be extended in two natural ways. First, from an information-theoretic viewpoint, all mutual information terms $I(S; Z)$ and $I(Y; Z \mid S)$ are defined for continuous $S$, and our variational approximation still applies if we model $p(S \mid Z)$ with a suitable regression head (e.g., Gaussian output) instead of a binary classifier. Second, on the IPW side, the propensity model $e(i)$ can be replaced by a generalized propensity or density-ratio model over continuous $S$ conditioned on covariates $x_i$, which is standard in causal inference with continuous treatments. In settings where evaluation uses discrete groups, one can also discretize or bin the continuous sensitive variable into intervals and apply the same implementation as in the binary case.

---

> ### Comment · Reviewer_xSKV · 2025-11-21
>
> I thank the authors for their feedback including the case studies - some follow up questions about the mentioned weaknesses and questions that concern me the most below.
>
> 1 - About the efficiency, it is highly recommended to involve running time and complexity analysis in the paper to provide a comprehensive understanding about the effiicency of the proposed method. This also closely relates to Q1.
>
> 2 - To meet the goal of modeling high order neighborhood effects in a stable and efficient way, other models such as SGC and GCNII can also be adopted. For example, GCNII is also used for capturing long range dependencies without the oversmoothing issues in deep GCN-style stacking. Can you provide further justification on the provided choice?

---

> > ### Author Response · Authors · 2025-11-26
> >
> > **Response to Q1**:
> >
> > We have incorporated a theoretical complexity analysis and updated empirical running-time results in the revised manuscript. We also revised several imprecise or inappropriate expressions. In terms of time complexity, although FairMIB introduces three shallow encoders that operate on the same graph, each encoder still follows a standard message-passing mechanism, so the computational cost remains linear in the number of edges and the hidden dimension, incurring only a moderate constant-factor overhead compared with single-view GNNs. Empirically, on the Bail dataset with 500 training epochs using an RTX 4060 (8GB) GPU, FairMIB takes 123 seconds, while NIFTY takes 83 seconds, FairGNN 97 seconds, EDITS 121 seconds, DAB-GNN 345 seconds, and FairVGNN as high as 580 seconds. These results indicate that, despite its multi-view design, FairMIB maintains strong computational efficiency in practice.
> >
> > **Response to Q2**:
> >
> > We appreciate your suggestion to evaluate alternative diffusion operators such as SGC and GCNII. We have  additional experiments on the Bail dataset comparing APPNP, SGC, and GCNII under the same setting (500 epochs on the same backbone and GPU). Using APPNP in the diffusion view achieves a good balance between utility, fairness, and efficiency (AUC 88.52 ± 1.45, F1 78.89 ± 1.91, DP 1.35 ± 1.23, EO 1.39 ± 0.68, time 123.5 s). SGC is slightly faster (115.2 s) but clearly worse in both utility and fairness (AUC 86.88 ± 1.92, F1 76.61 ± 2.27, DP 1.72 ± 1.45, EO 2.86 ± 2.01), indicating that its more aggressive simplification harms the diffusion view’s ability to support fair representation learning. GCNII attains the strongest utility and comparable fairness (AUC 89.09 ± 2.09, F1 80.27 ± 3.69, DP 1.31 ± 0.35, EO 1.39 ± 0.49), but at a noticeably higher computational cost (144.97 s) and with a substantially more complex architecture.
> >
> > Based on these results, we chose APPNP as a principled compromise: it captures long-range dependencies more effectively than SGC while remaining simpler and more efficient than GCNII, and it integrates cleanly with our IB-based formulation. Importantly, FairMIB is not tied to APPNP; the diffusion operator is modular, and GCNII or other propagation schemes can be plugged in as stronger but more expensive variants. We will report these ablation results and clarify this design choice in the revised manuscript.
> >
> > | **Setting / Method** | **AUC**      | **Accuracy** | **F1**       | **SP**      | **EO**      | **Time (s)** |
> > | -------------------- | ------------ | ------------ | ------------ | ----------- | ----------- | ------------ |
> > | APPNP                | 88.52 ± 1.45 | 84.48 ± 1.53 | 78.89 ± 1.91 | 1.35 ± 1.23 | 1.39 ± 0.68 | 123.4915     |
> > | GCN                  | 87.60 ± 2.11 | 83.50 ± 2.43 | 77.62 ± 3.11 | 1.53 ± 0.84 | 1.49 ± 0.66 | 130.2052     |
> > | SGC                  | 86.88 ± 1.92 | 84.03 ± 1.47 | 76.61 ± 2.27 | 1.72 ± 1.45 | 2.86 ± 2.01 | 115.1629     |
> > | GCNII                | 89.09 ± 2.09 | 85.49 ± 3.15 | 80.27 ± 3.69 | 1.31 ± 0.35 | 1.39 ± 0.49 | 144.9692     |
> > | Without IPW          | 88.26 ± 0.93 | 84.54 ± 1.46 | 78.91 ± 1.36 | 1.90 ± 0.75 | 1.72 ± 0.74 | 120.7191     |

---

### Official Review · Reviewer_BfRK · 2025-10-31

**Soundness:** 2
**Presentation:** 2
**Contribution:** 2
**Rating:** 2
**Confidence:** 3

**Summary:**

This paper proposes FairMIB, a fairness-aware framework for GNNs that decomposes graph information into feature, structural, and diffusion views. It applies a multi-view conditional information bottleneck (MCIB) objective to balance utility and fairness by minimizing mutual information with sensitive attributes while maximizing task-relevant information. Experiments on five benchmark datasets (German, Bail, Credit, Pokec-z, Pokec-n) show improved fairness metrics compared with existing baselines such as FairVGNN, GRAFair, and DAB-GNN.

**Strengths:**

1. The paper is clear and easy to follow.

2. They evaluate on five datasets and include several baseline comparisons and ablation studies, which gives some empirical support for the proposed approach.

**Weaknesses:**

1. The paper has limited novelty. Nearly every component of FairMIB, i.e., information bottleneck, multi-view consistency, contrastive learning, IPW correction, has been previously published. The contribution is primarily a combination of existing tools rather than a conceptual breakthrough. The paper’s framing as a “multi-view information bottleneck” for fairness appears incremental relative to GRAFair (Zhang et al. 2025) and FDGIB (Zheng et al. 2024). There is insufficient discussion of how FairMIB fundamentally advances beyond these prior works.

2. The paper claims to disentangle multiple sources of bias by constructing three graph views. However, there is no formal proof or empirical verification that the proposed decomposition actually isolates independent bias factors. In practice, node attributes, topology, and diffusion processes are highly entangled, treating them as separable “views” could be theoretically fragile and arguably artificial.

3. The advance over prior fairness models (e.g, GRAFair, DAB-GNN) is incremental at best, the paper provides little justification for its distinct contribution.

**Questions:**

1. How is the independence between views ensured or even approximately satisfied? If not independent, the theoretical motivation for a multi-view bottleneck collapses.

2. What is the true novelty over FDGIB and GRAFair? Beyond adding a diffusion view, FairMIB appears to remix existing elements.

3. In the ablation studies Figure 2 (b), for German and Bail datasets, why does removing the structure view lead to decreasing DP scores?

4. What is the runtime of FairMIB compared to baselines like FairVGNN and DAB-GNN?

5. Did you retune $\lambda$ and $\gamma$ in ablations? If not, results might unfairly penalize variants.

---

> ### Author Response · Authors · 2025-11-20
> **Clarifications on FairMIB’s Novelty, Multi-View Bias Modeling, and Empirical Analysis**
>
> We thank the reviewer for his time and valuable comments on our work.
>
>
>
> **Response to W1**:
>
> We would like to clarify that our goal is not to simply stack existing methodological components Instead, we aim to address a new problem paradigm: learning fair graph representations under explicitly multi-source bias arising from node features, graph structure, and diffusion patterns, all within an optimizable information, theoretic framework. In this setting, the modules in FairMIB serve specific, complementary roles, each completing information from a different view and aligning them into a shared fair latent space, rather than being combined in an ad hoc manner. FairMIB establishes a multi-view conditional information bottleneck (CIB) objective that (i) is theoretically grounded in IB and CFB principles, (ii) becomes trainable via variational lower bounds, and (iii) is instantiated with three views, feature, structural, and diffusion, jointly regularized by a contrastive multi-view consistency term. This yields a single, unified objective function that can be optimized end-to-end, instead of a heuristic sum of disconnected losses. Experiments demonstrate that this principled design consistently improves both utility and fairness. Compared with prior work, GRAFair applies the Conditional Fairness Bottleneck but operates on a single entangled view, without decomposing or completing heterogeneous sources of bias. FDGIB, though grounded in IB theory and leveraging causal disentanglement, still uses a single representation space and does not model cross-view alignment or view, specific bias propagation. In contrast, FairMIB introduces a multi-view bottleneck framework that explicitly completes feature, structural, and diffusion views and aligns them through consistency regularization, thereby advancing this research direction. We will expand this conceptual comparison and emphasize that FairMIB is specifically designed to address multi-source bias, which existing single-view IB-based methods are not equipped to handle.
>
>
>
> **Response to W2**:
>
>
>
> Thank you very much for your constructive comments. The motivation for constructing three graph views in FairMIB is to complement the information missing in each single view, so that the model can better capture bias from different perspectives and learn fairer representations.
>
>
>
> **Response to W3**：
>
> We respectfully argue that FairMIB is more than an incremental combination of existing ideas. Conceptually, it introduces a multi-view information bottleneck framework that explicitly models features, structure, and diffusion as complementary sources of bias and unifies them under a single optimizable objective that enforces both fairness and cross-view consistency. This differs substantially from GRAFair, which applies CFB on a single entangled view, and from DAB-GNN, which disentangles biases but does not employ IB-based multi-view bottlenecks or diffusion, guided debiasing. Empirically, FairMIB consistently achieves a stronger fairness, utility trade-off across datasets, supporting its distinct contribution in formulation and performance.

---

> ### Author Response · Authors · 2025-11-20
>
> **Response to Q1**：
>
>
>
> In our framework, the three views are not assumed to be independent, nor do we impose any independence constraints. Instead, the multi-view architecture is designed to complement missing information across views (features, structure, diffusion) so that each view contributes its own perspective on bias. The theoretical formulation extends the information bottleneck to a multi-view setting that remains valid without independence assumptions. The multi-view consistency term aims to align the partially correlated views into a shared fair representation, not to enforce independence.
>
>
>
> **Response to Q2**:
>
>
>
> We appreciate the reviewer's comments. I would like to clarify that the starting point of our approach is to address the missing information between views in order to reduce entangled bias. In FairMIB, node attributes, graph structure, and diffusion are treated as separable but not necessarily independent views. Our objective is not to enforce strict independence but to deconstruct biases within these views so that their influence can be better isolated and mitigated. The multi-view consistency loss is designed to ensure that each view promotes fairness without reinforcing bias, thus improving fairness while maintaining model utility. This multi-view bias deconstruction approach is conceptually similar to methods such as DAB-GNN and FairGKD. They both handle different aspects of graph data, such as attributes, structure, and potential biases, through separate modules to achieve debiasing. Specifically, DAB-GNN uses decoupling techniques to separate attributes, structure, and potential biases, and amplifies and debiases each type of bias individually. FairGKD addresses biases caused by node attributes and graph topology through distillation, while learning fair representations. Similarly, FairMIB handles multi-source bias through explicit information decomposition and complementary view construction, which is why we choose not to merge the views into a single entangled representation.
>
>
>
> **Response to Q3**:
>
> The phenomenon observed in the German and Bail datasets stems from how their graph structures are constructed. In these datasets, edges are created based on feature similarity rather than from real-world graph relations. As a result, the structural information can introduce additional bias, especially when sensitive attributes (e.g., gender, race) correlate with node features. When the structure view is removed, the model is no longer influenced by this synthetic structural bias and relies more on node features. This reduces structural bias and leads to lower DP scores (i.e., improved fairness). In contrast, the Pokec_c and Pokec_z datasets contain real-world graph structures based on actual social interactions, which carry more authentic relational patterns. Removing the structure view removes valuable structural signals and causes the model to rely excessively on biased attribute information, resulting in higher DP scores (i.e., reduced fairness). Thus, the contrasting results arise from the fundamental differences in graph construction across datasets.
>
>
>
> **Response to Q4**:
>
>
>
> Thank you for highlighting the need for a clearer runtime comparison. We conducted additional experiments under a unified setting: all models were trained for 500 epochs on the Bail dataset using the same GPU (RTX 4060, 8 GB). The training times are as follows: FairMIB requires 123 seconds, while NIFTY takes 83 seconds, FairGNN 97 seconds, EDITS 121 seconds, DAB-GNN 345 seconds, and FairVGNN 580 seconds. These results show that FairMIB is computationally efficient and achieves competitive fairness–utility trade-offs. Although FairMIB uses three encoders, this design does not cause a significant increase in training time in practice. We will include this runtime comparison in the revised manuscript.
>
> | **Metric**    | **FairMIB** | **NIFTY** | **FairGNN** | **EDITS** | **DAB-GNN** | **FairVGNN** |
> | ------------- | ----------- | --------- | ----------- | --------- | ----------- | ------------ |
> | Training time | 123s        | 83s       | 97s         | 121s      | 345s        | 580s         |
>
> **Response to Q5**:
>
> Thank you for raising this concern. For all ablation variants, we did retune the hyperparameters (including the trade-off coefficients such as $\lambda$ and $\gamma$ ) rather than reusing the settings from the full FairMIB model. Specifically, each variant was tuned via grid search on a validation set, and we report the best-performing configuration for that variant. Therefore, the ablation results do not suffer from unfair penalization due to suboptimal hyperparameter choices. We will clarify this implementation detail in the revised manuscript.

---

### Author Response · Authors · 2025-11-29

We thank all reviewers for their careful reading and constructive feedback. Below we briefly summarize the main clarifications and additional results provided in our rebuttal and discussion.

**Conceptual contribution and relation to prior work**

We clarified that FairMIB is not a simple stacking of existing modules, but a unified multi-view conditional information bottleneck framework tailored to multi-source bias in graphs. It jointly models feature, structural, and diffusion views as complementary, biased information sources and aligns them via a single optimizable CIB objective with a multi-view consistency regularizer. This is substantially different from single-view IB-based methods such as GRAFair and FDGIB, and from DAB-GNN/FairGKD, which do not employ an IB-based multi-view bottleneck or explicit diffusion-guided debiasing.

**Theoretical assumptions and use of views**

We clarified that FairMIB does not assume independence between views. The three views are treated as correlated but complementary; the objective remains valid without independence assumptions, and the consistency term is used to align partially correlated views into a shared fair representation rather than to enforce independence.

**Behavior on different datasets**

We explained the seemingly contradictory ablation results when removing the structural view: for German/Bail, the synthetic similarity-based graph introduces extra structural bias, so dropping structure improves DP; for Pokec\_c/z, the real social graph carries useful relational signals, so removing structure hurts fairness and utility.

**New empirical evidence on efficiency and design choices**

We added controlled runtime experiments on Bail (500 epochs, same GPU). FairMIB is only moderately slower than the lightest baselines (e.g., 123s vs. 83s/97s for NIFTY/FairGNN) and much faster than heavier models (345s/580s for DAB-GNN/FairVGNN). Ablations on diffusion operators (GCN, APPNP, SGC, GCNII) and IPW show that APPNP offers a good balance of fairness--utility--efficiency and that IPW significantly improves fairness without hurting utility. All ablation variants were independently retuned.

**Practicality, extensibility, and implementation details**

We clarified how IPW is implemented (a small MLP on non-sensitive features with stabilized clipping), and discussed how the framework naturally extends to multiple and continuous sensitive attributes, as well as to partially missing or noisy sensitive labels, without changing the core IB formulation.

---

### Meta-Review · Area_Chair_usAQ · 2025-12-09

**Summary:**

(*Disclaimer: given the peculiar review process, some of my choices and reasonings below will be highly subjective, as I tried to imagine how a reviewer would have reacted to a specific response. I understand that any negative choice will be perceived as unfair by the authors, and I apologize in advance for that.*)

The paper describes a contrastive learning framework for training fair GNNs. The core idea is to consider multiple views of the same graph (three of them: feature-only, graph-only, and a "diffused" view), and then apply a mutual information criterion to learn a discriminative representation which is also fair. Experiments support the method on several benchmarks.

Of the three reviews, one is highly positive (`Wuzn`), but it is also (in my opinion) the most "shallow" in terms of analysis. The other two reviewers (`xSKV`, `Wuzn`) are more critical, and they focus, among others, on limited novelty, lack of computational cost analysis, limitations in the methodology, and writing quality.

As I argue below, the authors provided a mixed rebuttal. Some concerns were easily addressed (e.g., training time), while other answers (e.g., limited novelty or theoretical motivation) do not seem convincing. For example, there is ample literature on multi-view / contrastive / fair GNNs, and the concern that the method is a generic combination of (empirically motivated) components seems valid. I do not think the rebuttal would have moved the two low scores above a weak rejection.

**Reviewer Concerns:**

1) **Computational cost** (`Wuzn`, `xSKV`, `BfRK`): the authors added a big-O analysis to the paper and a runtime experiment on a single dataset. While the experiment could have been enlarged to other datasets, I think this issue would have been considered solved by most reviewers.

2) **Propensity scores** (`Wuzn`): I think enough details were added to address this concern.

3) **Choice of APPNP** (`xSKV`): this is hard to evaluate, because the answer boils down to "it works better on this ablation study compared to a standard GNN diffusion", which could have been received either well or not. I personally find this choice less critical than the views (see below), so I gave it less impact in my decision.

4) **Writing quality** (`xSKV`): while the writing is heavy at time, the paper does not seem so inconsistent in writing (there are only minor issues, such as citing format or few typos, such as "DIFFSION"). Putting methodological details in the appendix seems a common practice in conferences today (where more space is given to the experiments), and it is generally a subjective choice. My only concern here is appendix "C.1 BALANCE DIFFSION VIEW", which is extremely short and lacks important details.

5) **Limitation to binary sensitive values** (`xSKV`): the authors describe how the framework could have been extended, but they do not provide an experimental evaluation. While (once again) this is subjective, I believe this could have been a strong negative factor for the final evaluation.

6) **Novelty / validity** (`BfRK`): this is the strongest concern across the three reviewers, suggesting the paper is only a "patchwork" of existing components and (more importantly) lacking a good theoretical motivation on the choice of the three views. I found the answer quite generic and not addressing the issues in depth. This was a major factor in my final decision.

7) **Hyper-parameter tuning for the ablations** (`BfRK`): this was clarified in the rebuttal.

**Reviewer Scores:**

`Wuzn`: they start from a very high score (8), and both questions were addressed well. I believe the score would have remained unchanged.

`xSKV`: this is very subjective, since point 1 was clearly addressed, while points 3,4,5 are more subjective. An additional discussion could have gone both ways. At most, the reviewer would have increased the score to 6.

`BfRK`: as I argued above, I do not believe point 6 was addressed. I do not think the reviewer would have viable to going beyond 4 (at most).

Overall, the most positive scenario leaves the paper with a highly positive reviewer, a borderline one, and a highly critical one.

---

### Decision · Program_Chairs · 2026-01-26

Reject